# DISTRIBUTIONAL BELLMAN OPERATORS OVER MEAN EMBEDDINGS

## ABSTRACT

We propose a novel algorithmic framework for distributional reinforcement learning, based on learning finite-dimensional mean embeddings of return distributions. We derive several new algorithms for dynamic programming and temporal-difference learning based on this framework, provide asymptotic convergence theory, and examine the empirical performance of the algorithms on a suite of tabular tasks. Further, we show that this approach can be straightforwardly combined with deep reinforcement learning, and obtain a new deep RL agent that improves over baseline distributional approaches on the Arcade Learning Environment.

## 1 INTRODUCTION

In distributional approaches to reinforcement learning (RL), the aim is to learn the full probability distribution of future returns (Morimura et al., 2010a; Bellemare et al., 2017; 2023), rather than just their expected value, as is typically the case in value-based reinforcement learning (Sutton & Barto, 2018). Distributional RL was proposed in the setting of deep reinforcement learning by Bellemare et al. (2017), with a variety of precursor work stretching back almost as far as Markov decision processes themselves (Jaquette, 1973; Sobel, 1982; Chung & Sobel, 1987; Morimura et al., 2010a;b). Beginning with the work in Bellemare et al. (2017), the distributional approach to reinforcement learning has been central across a variety of applications of deep RL in simulation and in the real world (Bodnar et al., 2020; Bellemare et al., 2020; Wurman et al., 2022; Fawzi et al., 2022).

Typically, predictions of return distributions are represented directly as approximate probability distributions, such as categorical distributions (Bellemare et al., 2017). Rowland et al. (2019) proposed an alternative framework where return distributions are represented via the values of *statistical functionals*, called a *sketch* by Bellemare et al. (2023). This provided a new space of distributional reinforcement learning algorithms, leading to improvements in deep RL agents, and hypotheses regarding distributional RL in the brain (Dabney et al., 2020; Lowet et al., 2020). On the other hand, a potential drawback of this approach is that each distributional Bellman update to the representation, these values must be "decoded" back into an approximate distribution via an *imputation strategy*. In practice, this can introduce significant computational overhead to Bellman updates, and is unlikely to be biologically plausible for distributional learning in the brain (Tano et al., 2020).

Here, we focus on a notable instance of the sketch called the *mean embedding sketch*. In short, the mean embedding is the expectation of nonlinear functions under the distribution represented (Smola et al., 2007; Sriperumbudur et al., 2010; Berlinet & Thomas-Agnan, 2011), and is related to *frames* in signal processing (Mallat, 1999) and *distributed distributional code* in neuroscience (Sahani & Dayan, 2003; Vértes & Sahani, 2018). The core contributions of this paper are to revisit the approach to distributional reinforcement learning based on sketches (Rowland et al., 2019), and to propose the *sketch Bellman operator* that updates the implicit distributional representation as a simple linear operation, obviating the need for the expensive imputation strategies converting between sketches and distributions. This provides a rich new space of distributional RL algorithms that operate entirely in the space of sketches. We provide theoretical convergence analysis to accompany the framework, investigate the practical behaviour of various instantiations of the proposed algorithms in tabular domains, and demonstrate the effectiveness of the sketch framework in deep reinforcement learning, showing that our approach is robust enough to serve as the basis for a new variety of deep distributional reinforcement learning algorithms.

## 2 BACKGROUND

We consider a Markov decision process (MDP) with state space $\mathcal{X}$, action space $\mathcal{A}$, transition probabilities $P : \mathcal{X} \times \mathcal{A} \to \mathscr{P}(\mathcal{X})$, reward distribution function $P_R : \mathcal{X} \times \mathcal{A} \to \mathscr{P}(\mathbb{R})$, and discount factor $\gamma \in [0, 1)$. Given a policy $\pi : \mathcal{X} \to \mathscr{P}(\mathcal{A})$ and initial state $x \in \mathcal{X}$, a random trajectory $(X_t, A_t, R_t)_{t \geq 0}$ is the sequence of random states, actions, and rewards encountered when using the policy $\pi$ to select actions in this MDP. More precisely, we have $X_0 = x$, $A_t \sim \pi(\cdot|X_t)$, $R_t \sim P_R(X_t, A_t)$, $X_{t+1} \sim P(\cdot|X_t, A_t)$ for all $t \geq 0$. We write $\mathbb{P}_x^\pi$ and $\mathbb{E}_x^\pi$ for probabilities and expectations with respect to this distribution, respectively. The performance along the trajectory is measured by the discounted return, defined by

$$\sum_{t=0}^{\infty} \gamma^t R_t. \tag{1}$$

In typical value-based reinforcement learning, during policy evaluation, the agent learns the expectation of the return for each possible initial state $x \in \mathcal{X}$, which is encoded by the value function $V^\pi : \mathcal{X} \to \mathbb{R}$, given by $V^\pi(x) = \mathbb{E}_x^\pi[\sum_{t=0}^{\infty} \gamma^t R_t]$.

### 2.1 DISTRIBUTIONAL RL AND THE DISTRIBUTIONAL BELLMAN EQUATION

In distributional reinforcement learning, the problem of policy evaluation is to learn the probability distribution of return in Equation (1) for each possible initial state $x \in \mathcal{X}$. This is encoded by the return-distribution function $\eta^\pi : \mathcal{X} \to \mathscr{P}(\mathbb{R})$, which maps each initial state $x \in \mathcal{X}$ to the corresponding distribution of the random return, e.g. $\eta^\pi(x)$ is the return distribution of state $x$. A central result in distributional reinforcement learning is the distributional Bellman equation, which relates the distribution of the random return under different combinations of initial states and actions.

To build the random variable formulation of the returns, we let $(G^\pi(x) : x \in \mathcal{X})$ be a collection of random variables with the property that $G^\pi(x)$ is equal to Equation (1) in distribution, conditioned on the initial state $X_0 = x$. This formulation implies that the random variable $G^\pi(x)$ is distributed as $\eta^\pi(x)$, introduced above, for all $x \in \mathcal{X}$. Consider a random transition $(x, R, X')$ generated by $\pi$, independent of the $G^\pi$ random variables. Then, the (random variable) distributional Bellman equation states that for each initial state $x$,

$$G^\pi(x) \overset{\mathcal{D}}{=} R + \gamma G^\pi(X') \quad | X = x.$$

Here, we use the slight abuse of the conditioning bar to set the distribution of $X$ in the random transition. It is also useful to introduce the distributional Bellman operator $\mathcal{T}^\pi : \mathscr{P}(\mathbb{R})^{\mathcal{X}} \to \mathscr{P}(\mathbb{R})^{\mathcal{X}}$ to describe the transformation that occurs on the right-hans side (Morimura et al., 2010a; Bellemare et al., 2017). If $\eta \in \mathscr{P}(\mathbb{R})^{\mathcal{X}}$ is a collection of probability distributions, and $(G(x) : x \in \mathcal{X})$ is a collection of random variables such that $G(x) \sim \eta(x)$ for all $x$, and $(X, R, X')$ is random transition generated by $\pi$, independent of $(G(x) : x \in \mathcal{X})$, then $(\mathcal{T}^\pi \eta)(x) = \mathrm{Dist}(R + \gamma G(X')|X = x)$.

To implement algorithms of distributional RL, one needs to approximate the infinite-dimensional return-distribution function $\eta^\pi$ with finite-dimensional representations. This is typically done via direct approximations in the space of distributions; see e.g. Bellemare et al. (Chapter 5; 2023).

### 2.2 STATISTICAL FUNCTIONALS AND SKETCHES

Rather than using approximations in the space of distributions, Rowland et al. (2019) proposed to represent return distributions indirectly via *functionals* of the return distribution, called *sketches* by Bellemare et al. (2023). In this work we consider a specific class of sketches, defined below.

**Definition 2.1** (Mean embedding sketches). A *mean embedding sketch* $\psi$ is specified by a function $\phi : \mathbb{R} \to \mathbb{R}^m$, and defined by

$$\psi(\nu) := \mathbb{E}_{Z \sim \nu}[\phi(Z)]. \tag{2}$$

For a given distribution $\nu$, the embedding $\psi(\nu)$ can therefore be thought of as providing a *lossy* summary of the distribution. The name here is motivated by the kernel literature, in which Equation (2) can be viewed as embedding the distribution $\nu$ into $\mathbb{R}^m$ based on the mean value of $\phi$ under this distribution (Smola et al., 2007; Sriperumbudur et al., 2010; Berlinet & Thomas-Agnan, 2011).

Statistical functional dynamic programming and temporal-difference learning (SFDP/SFTD; Rowland et al. (2019), see also Bellemare et al. (2023)) is an approach to distributional RL in which sketch values, rather than approximate distributions, are the primary object learned. Given a sketch $\psi$ and estimated sketch values $U : \mathcal{X} \to \mathbb{R}^m$, these approaches proceed by first defining an *imputation strategy* $\iota : \mathbb{R}^m \to \mathscr{P}(\mathbb{R})$ mapping sketch values back to distributions, with the aim that $\psi(\iota(U)) \approx U$, so that $\iota$ behaves as an approximate pseudo-inverse to $\psi$. The usual Bellman backup is then applied to this *imputed* distribution, and the sketch value extracted from this updated distribution. Thus, if $U : \mathcal{X} \to \mathbb{R}^m$ represents approximations to sketch values, a typical update in SFDP takes the form $U \leftarrow \psi((\mathcal{T}^\pi \iota(U))(x))$ (Bellemare et al., 2023); see Figure 1.

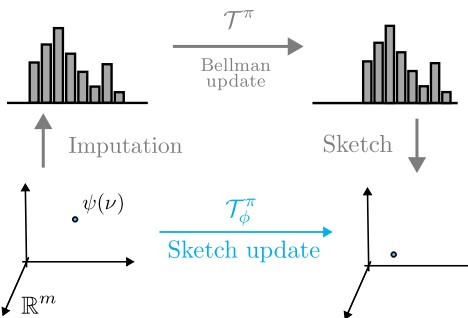

Figure 1: The statistical functional framework proposed by Rowland et al. (2019) (top), and the framework proposed in this paper, directly updating sketches, avoiding the imputation step (bottom).

This approach led to expectile-regression DQN, a deep RL agent that aims to learn the sketch values associated with certain expectiles (Newey & Powell, 1987) of the return, and influenced a distributional model of dopamine signalling in the brain (Dabney et al., 2020). An important consideration is that computation of the imputation strategy is often costly in machine learning applications, and considered biologically implausible in neuroscience models (Tano et al., 2020).

## 3 THE BELLMAN SKETCH FRAMEWORK

Our goal is to derive a framework for approximate computation of the sketch $\psi$ (with corresponding feature function $\phi$) of the return distributions corresponding to a policy $\pi$, without needing to design, implement, and compute an imputation strategy as in the case of SFDP/TD; see Figure 1 for a visual comparison of the two approaches. That is, we aim to compute the object $U^\pi : \mathcal{X} \to \mathbb{R}^m$, given by

$$U^\pi(x) := \psi(\eta^\pi(x)) = \mathbb{E}^\pi_x[\phi(\textstyle\sum_{t=0}^\infty \gamma^t R_t)].$$

We begin by considering environments with a finite set of possible rewards $\mathcal{R} \subseteq \mathbb{R}$; we discuss generalisations later. To motivate our method, we first consider a special case; suppose that for each possible return $g \in \mathbb{R}$, and each possible immediate reward $r \in \mathcal{R}$, there exists a matrix $B_r$ such that

$$\phi(r + \gamma g) = B_r \phi(g); \tag{3}$$

note that $B_r$ does not depend on $g$, and $\gamma$ is a constant. In words, this says that the feature function $\phi$ evaluated at the bootstrap return $r + \gamma g$ is expressible as a linear transformation of the feature function evaluated at $g$ itself. If such a relationship holds, then we have

$$U^\pi(x) \stackrel{(a)}{=} \mathbb{E}^\pi_x[\phi(R + \gamma G^\pi(X'))] \stackrel{(b)}{=} \mathbb{E}^\pi_x[B_R \phi(G^\pi(X'))] \stackrel{(c)}{=} \mathbb{E}^\pi_x[B_R U^\pi(X')], \tag{4}$$

where (a) follows from the distributional Bellman equation, (b) follows from Equation (3), and (c) from exchanging the linear map $B_r$ and the conditional expectation given $(R, X')$, crucially relying on the linearity of the approximation in Equation (3). Note that for example with $\phi(g) = (1, g)^\top$ we have $B_r = \begin{pmatrix} 1 & 0 \\ r & \gamma \end{pmatrix}$, and Equation (4) reduces to the classical Bellman equation for $V^\pi$, with $U^\pi(x) = (1, V^\pi(x))^\top$.

Thus, $U^\pi(x)$ satisfies its own linear Bellman equation, which motivates algorithms that work directly in the space of sketches, without recourse to imputation strategies. In particular, a natural dynamic programming algorithm to consider is based on the recursion

$$U(x) \leftarrow \mathbb{E}^\pi_x[B_R U(X')]. \tag{Sketch-DP}$$

As this is an update applied directly to sketch values themselves, we introduce the *sketch Bellman operator* $\mathcal{T}^\pi_\phi : (\mathbb{R}^m)^{\mathcal{X}} \to (\mathbb{R}^m)^{\mathcal{X}}$, with $(\mathcal{T}^\pi_\phi U)(x)$ defined according to the right-hand side of

Equation (Sketch-DP). Note that $\mathcal{T}_\phi^\pi$ is a *linear* operator, in contrast to the standard expected-value Bellman operator, which is affine. We recover the affine case by taking one component of $\phi$ to be constant, e.g. $\phi_1(g) \equiv 1$, and enforcing $U_1(x) \equiv 1$.

The right-hand side of Equation (Sketch-DP) can be unbiasedly approximated with a sample transition $(x, r, x')$. Stochastic approximation theory (Kushner & Yin, 1997; Bertsekas & Tsitsiklis, 1996) then naturally suggests the following temporal-difference learning update, given a learning rate $\alpha$:

$$U(x) \ \leftarrow \ (1 - \alpha)U(x) + \alpha B_r V(x'), \qquad \text{(Sketch-TD)}$$

Rowland et al. (2019) introduced the term *Bellman closed* for sketches for which an *exact* dynamic programming algorithm is available, and provided a characterisation of Bellman closed mean embedding sketches. The notion of Bellman closedness is closely related to the relationship in Equation (3), and from Rowland et al. (Theorem 4.3; 2019), we can deduce that the only mean embedding sketches that satisfy Equation (3) are invertible linear combinations of first-$m$ moments.

Thus, our discussion above serves as a way of re-deriving known algorithms for computing moments of the return (Sobel, 1982; Lattimore & Hutter, 2014), but is insufficient to yield algorithms for computing other sketches. Additionally, since moments of the return distribution are naturally of widely differing magnitudes, it is difficult to learn a high-dimensional mean embedding based on moments; see Appendix D.3 for further details. To go further, we must weaken the assumption made in Equation (3).

## 3.1 GENERAL SKETCHES

To extend our framework to a much more general family of sketches, we relax our assumption of the exact predictability of $\phi(r + \gamma g)$ from $\phi(g)$ in Equation (3), by defining a matrix of *Bellman coefficients* $B_r$ for each possible reward $r \in \mathcal{R}$ as the solution of the linear regression problem:

$$B_r \ := \ \arg\min_B \mathbb{E}_{G \sim \mu}\left[\|\phi(r + \gamma G) - B\phi(G)\|_2^2\right], \qquad (5)$$

so that, informally, we have $\phi(r + \gamma g) \approx B_r \phi(g)$ for each $g$. Here, $\mu$ is a distribution to be specified that weights the returns $G$. Using the same motivation as in the previous section, we therefore obtain

$$U^\pi(x) \ \overset{(a)}{=} \ \mathbb{E}_x^\pi[\phi(R + \gamma G^\pi(X'))] \ \approx \ \mathbb{E}_x^\pi[B_R \phi(G^\pi(X'))] \ \overset{(c)}{=} \ \mathbb{E}_x^\pi[B_R U^\pi(X')], \qquad (6)$$

noting that informally we have *approximate* equality in the middle of this line. This still motivates the approaches expressed in Equations (Sketch-DP) and (Sketch-TD), though we have lost the property that the exact sketch values $U^\pi$ are a fixed point of the dynamic programming procedure.

**Computing Bellman coefficients.** Under mild conditions (invertibility of $C$ as follows) the matrix of Bellman coefficients $B_r$ defined in Equation (5) can be expressed as $B_r = C_r C^{-1}$, where $C, C_r \in \mathbb{R}^{m \times m}$ are defined by

$$C := \mathbb{E}_{G \sim \mu}[\phi(G)\phi(G)^\top], \qquad (7)$$
$$C_r := \mathbb{E}_{G \sim \mu}[\phi(r + \gamma G)\phi(G)^\top].$$

The elements of these matrices are expressible as integrals over the real line, and hence several possibilities are available for (approximate) computation: if $\mu$ is finitely-supported, direct summation is possible; in certain cases the integrals may be analytically available, and otherwise numerical integration can be performed. Additionally, for certain feature maps $\phi$, the Bellman coefficients $B_r$ have particular structure that can be exploited computationally; see Appendix B.3 for further discussion. Detailed properties of $B_r$ are studied in Appendix B.5.

---

**Algorithm 1** Sketch-DP/Sketch-TD

---

\# Precompute Bellman coefficients
Compute $C$ as in Equation (7)
**for** $r \in \mathcal{R}$ **do**
  Compute $C_r$ as in Equation (7)
  Set $B_r = C_r C^{-1}$
**end for**
Initialise $U : \mathcal{X} \to \mathbb{R}^m$
\# Main loop
**if** DP **then**
  **for** $k = 1, 2, \ldots$ **do**
    $U(x) \leftarrow \sum_{r,x',a} P(r, x'|x, a)\pi(a|x)B_r U(x') \ \forall x$
  **end for**
**else if** TD **then**
  **for** $k = 1, 2, \ldots$ **do**
    Observe transition $(x_k, a_k, r_k, x_k')$.
    $U(x_k) \leftarrow (1 - \alpha_k)U(x_k) + \alpha_k B_{r_k} U(x_k')$
  **end for**
**end if**

---

**Algorithms.** We summarise the two core algorithmic contributions, **sketch dynamic programming** (Sketch-DP) and **sketch temporal-difference learning** (Sketch-TD), that arise from our proposed framework in Algorithm 1. Pausing to take stock, we have proposed an algorithm framework for computing approximations of *lossy* mean embeddings for a wide variety of feature functions $\phi$. Further, these algorithms operate directly within the space of sketch values.

**Selecting feature maps.** A natural question is what effects the choice of feature map $\phi$ has on the performance of the algorithm. There are several competing concerns. First, the richer the map $\phi$, the more information about the return distribution can be captured by the corresponding mean embedding. However, the computational costs (both in time and memory) of our proposed algorithms scale in the worst case cubically with $m$, the dimensionality of the mean embedding. In addition, the accuracy of the algorithm in approximating the mean embeddings of the true return distributions relies on having a low approximation error in Equation (6), which in turn relies on a low regression error in Equation (5) (see Proposition 4.1 below). Selecting an appropriate feature map is therefore somewhat nuanced, and involves trading off a variety of computational and approximation concerns.

A collection of feature maps we will use throughout the paper that offer the potential for trade-offs along the dimensions identified above is given by the translation family

$$\phi_i(z) := \kappa(s(z - z_i)), \ \ \forall\, i \in \{1, \cdots, m\}, \tag{8}$$

where $\kappa : \mathbb{R} \to \mathbb{R}$ is a *base feature function*, $s \in \mathbb{R}^+$ is the *slope*, and the set $\{z_1, \ldots, z_m\} \subseteq \mathbb{R}$ is the *anchors* of the feature map. We will often take $\kappa$ to be commonly used bounded and smooth nonlinear functions, such as the Gaussian or the sigmoid functions, and spread the anchor points over the return range. We emphasise that in principle there are no restrictions on the feature maps that can be considered in the framework; see Appendix B.2 for other possible choices.

**Remark 3.1** (Invariance). Given the $m$-dimensional function space obtained from the span of the coordinate functions $\phi_1, \ldots, \phi_m$, the algorithms proposed above are essentially independent of the choice of basis for this space. For any invertible matrix $M \in \mathbb{R}^{m \times m}$, replacing $\phi$ by $M^{-1}\phi$, and *also* $||\cdot||_2$ by $||\cdot||_{M^\top M}$ in Equation (5) gives an equivalent algorithm.

**Remark 3.2** (The need for *linear* regression). It is tempting to try and obtain a more general framework by allowing *non-linear* regression of $\phi(r + \gamma g)$ on $\phi(g)$ in Equation (5), to obtain a more accurate fit, for example fitting a function $H : \mathbb{R} \times \mathbb{R}^m \to \mathbb{R}^m$ so that $\phi(r + \gamma g) \approx H(r, \phi(g))$. The issue is that if $H$ is not linear in the second argument, then generally $\mathbb{E}[H(r, \phi(G(X')))] \neq H(r, \mathbb{E}[\phi(G(X'))])$, and so step (c) in Equation (6) is not valid. However, there may be settings where it is desirable to *learn* such a function $H$, to avoid online computation of Bellman coefficients every time a new reward is encountered in TD learning.

### 3.2 SKETCH-DP AT WORK

To provide more intuition for the Bellman sketch framework, we provide a walk-through of using Algorithm 1 to estimate the return distributions for the environment in Figure 2A; full details for replication are given in Appendix C. We take a feature map $\phi$ of the form given in Equation (8), taking $\kappa$ to be the sigmoid function, and $m = 13$ anchors evenly spaced between $-4.5$ and $4.5$ (Figure 2B). The Bellman regression problem in Equation (5) is set with $\mu = \text{Uniform}([-4, 4])$, based on the typical returns observed in the environment. The anchors and the choice of $\mu$ for regression are important but can be set following simple heuristics; see Appendices B.2 and B.3. We then run the Sketch-DP algorithm with the initial estimates $U(x)$ set to $\phi(0)$ for all $x \in \mathcal{X}$.

We compare the estimates produced by Sketch-DP against ground-truth by estimating the true mean embeddings from a large number of Monte Carlo samples of the returns from each state. Figure 2C (top) shows illustrates the ground-truth embeddings for each state, and Figure 2D (top) compares these ground-truth embeddings with those computed by Sketch-DP as the algorithm progresses; by 30 iterations, the mean embeddings are very close to the ground-truth.

To aid interpretation of these results, we also include a comparison in which we "decode" the mean embeddings back into probability distributions (via an imputation strategy (Rowland et al., 2019)), and compare with the ground-truth return distributions, projected onto the anchor locations of the features (Rowland et al., 2018). Full details of the imputation strategy are in Appendix B.1. These results are shown in the bottom panels of Figure 2C & D. Initially, the imputed distributions of the Sketch-DP mean embedding estimates reflect the initialisation to the mean embedding of $\delta_0$,

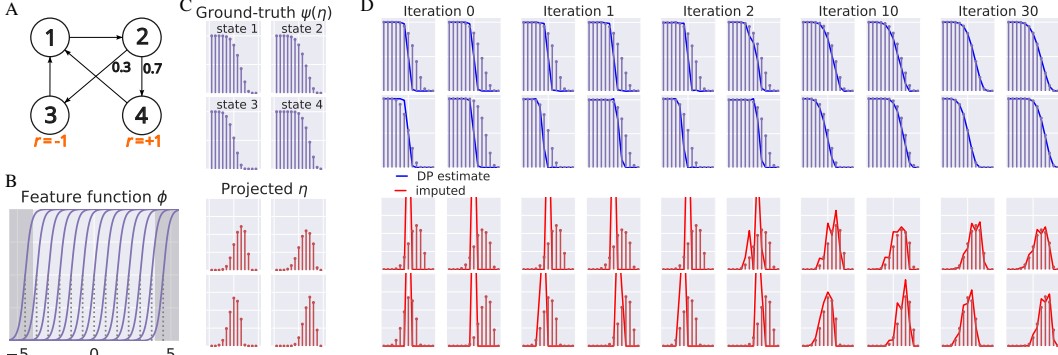

Figure 2: A: State transitions and rewards in the environment. B: The feature functions $\phi$ for the sketch-DP. Dotted lines indicate anchors. The regression Equation (5) is performed under a densely spaced grid over the light region $[-4, 4]$. C: The ground-truth mean embeddings under the sigmoid features in B, and the categorical projection of the ground-truth distribution onto the anchors of $\phi$ in B. D: The evolution of the estimated mean embeddings (bright blue lines) and imputed distributions (bright red lines) during Sketch-DP. The stems are the respective ground-truth from panel C.

though as more iterations of Sketch-DP are applied, the imputed distributions become close to the ground-truth. This indicates that, in this example, not only does Sketch-DP compute accurate mean embeddings of the return, but that this embedding is rich enough to recover a lot of information regarding the return distributions themselves.

Concluding the introduction of the Sketch-DP algorithmic framework, there are several natural questions that arise. Can we quantify how accurately Sketch-DP algorithms can approximate mean embeddings of return distributions? What effects do choices such as the feature map $\phi$ have on the algorithms in practice? The next sections are devoted to answering these questions in turn.

## 4 CONVERGENCE ANALYSIS

We analyse the Sketch-DP procedure described in Algorithm 1, which can be mathematically described in the following succinct manner. We let $U_0 : \mathcal{X} \to \mathbb{R}^m$ denote the initial sketch value estimates, and then note from Algorithm 1 that the collection of estimates after each DP update form a sequence $(U_k)_{k=0}^{\infty}$, with $U_{k+1} = \mathcal{T}_\phi^\pi U_k$. Our convergence analysis therefore focuses on the asymptotic behaviour of this sequence. We introduce the notation $\Phi : \mathscr{P}(\mathbb{R}) \to \mathbb{R}^m$ for the sketch associated with the feature function $\phi$, so that $\Phi\mu = \mathbb{E}_{Z\sim\mu}[\phi(Z)]$, and define $\Phi$ for return-distribution functions by specifying for $\eta \in \mathscr{P}(\mathbb{R})^{\mathcal{X}}$ that $(\Phi\eta)(x) = \Phi(\eta(x))$. Ideally, we would

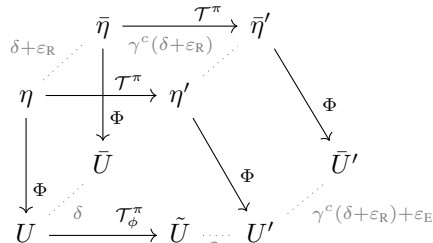

Figure 3: The objects and structure used to analyse the Sketch-DP algorithm.

like these iterates to approach $U^\pi : \mathcal{X} \to \mathbb{R}^m$, the sketch values of the true return distributions, given by $U^\pi(x) = \mathbb{E}_x^\pi[\phi(\sum_{t=0}^{\infty} \gamma^t R_t)]$. As already described, typically this is not possible when the sketch $\Phi$ is not Bellman closed, and so we can only expect to approximate $U^\pi$. Mathematically, this is because in general we have $\Phi\mathcal{T}^\pi \neq \mathcal{T}_\phi^\pi\Phi$ when $\phi$ is not Bellman closed.

The first step is to bound the error incurred in a single step of dynamic programming due to using $\mathcal{T}_\phi^\pi$ directly on the sketch values, rather taking sketch values after applying the true distributional Bellman operator to the underlying distributions; this corresponds to the foreground of Figure 3.

**Proposition 4.1. (Regression error to Bellman approximation.)** Let $\|\cdot\|$ be a norm on $\mathbb{R}^m$. Then for any return-distribution function $\eta \in \mathscr{P}([G_{\min}, G_{\max}])^{\mathcal{X}}$, we have

$$\max_{x\in\mathcal{X}} \|\Phi(\mathcal{T}^\pi\eta)(x) - (\mathcal{T}_\phi^\pi\Phi\eta)(x)\| \leq \sup_{g\in[G_{\min}, G_{\max}]} \max_{r\in\mathcal{R}} \|\phi(r + \gamma g) - B_r\phi(g)\|. \tag{9}$$

The second step of the analysis is to chain together the errors that are incurred at each step of dynamic programming, so as to obtain a bound on the asymptotic distance of the sequence $(U_k)_{k=0}^{\infty}$

from $U^\pi$, motivated by error propagation analysis in the case of function approximation (Bertsekas & Tsitsiklis (1996); Munos (2003); see also Wu et al. (2023) in the distributional setting). The next proposition provides the technical tools required for this; the notation is chosen to match the illustration in Figure 3.

**Proposition 4.2.** (**Error propagation.**) Consider a norm $\|\cdot\|$ on $\mathbb{R}^m$, and let $\|\cdot\|_\infty$ be the norm on $(\mathbb{R}^m)^{\mathcal{X}}$ defined by $\|U\|_\infty = \max_{x\in\mathcal{X}} \|U(x)\|$. Let $d$ be a metric on return-distribution functions (RDFs) such that $\mathcal{T}^\pi$ is a $\gamma^c$-contraction with respect to $d$. Suppose the following bounds hold.

- (Bellman approximation bound.) For any $\eta \in \mathscr{P}([G_{\min}, G_{\max}])^{\mathcal{X}}$,

$$\max_{x\in\mathcal{X}} \|\Phi(\mathcal{T}^\pi\eta)(x) - (\mathcal{T}_\phi^\pi\Phi\eta)(x)\| \leq \varepsilon_{\mathrm{B}}.$$

- (Reconstruction error bound.) For any $\eta, \bar{\eta} \in \mathscr{P}([G_{\min}, G_{\max}])^{\mathcal{X}}$ with sketches $U, \bar{U}$, we have $d(\eta, \bar{\eta}) \leq \|U - \bar{U}\|_\infty + \varepsilon_{\mathrm{R}}$.

- (Embedding error bound.) For any $\eta', \bar{\eta}' \in \mathscr{P}([G_{\min}, G_{\max}])^{\mathcal{X}}$ with sketches $U', \bar{U}'$, we have $\|U' - \bar{U}'\|_\infty \leq d(\eta', \bar{\eta}') + \varepsilon_{\mathrm{E}}$.

Then for any two return-distribution functions $\eta, \bar{\eta} \in \mathscr{P}([G_{\min}, G_{\max}])^{\mathcal{X}}$ with sketches $U, \bar{U}$ satisfying $\|U - \bar{U}\| \leq \delta$, we have

$$\|\Phi\mathcal{T}^\pi\eta - \mathcal{T}_\phi^\pi\bar{U}\|_\infty \leq \gamma^c(\delta + \varepsilon_{\mathrm{R}}) + \varepsilon_{\mathrm{R}} + \varepsilon_{\mathrm{E}}.$$

A formal proof is given in Appendix A; Figure 3 (bottom) shows the intuition, propagating bounds through different intermediate stages of the analysis of the update. We now state the main error bound result, which combines the two earlier results.

**Proposition 4.3.** Suppose the assumptions of Proposition 4.2 hold, that $\mathcal{T}^\pi$ maps $\mathscr{P}([G_{\min}, G_{\max}])^{\mathcal{X}}$ to itself, and suppose $\mathcal{T}_\phi^\pi$ maps $\{\Phi\nu : \nu \in \mathscr{P}([G_{\min}, G_{\max}])^{\mathcal{X}}\}$ to itself. Then for a sequence of sketches $(U_k)_{k=0}^\infty$ defined iteratively via $U_{k+1} = \mathcal{T}_\phi^\pi U_k$, we have

$$\limsup_{k\to\infty} \|U_k - U^\pi\| \leq \frac{1}{1-\gamma^c}(\gamma^c\varepsilon_{\mathrm{R}} + \varepsilon_{\mathrm{B}} + \varepsilon_{\mathrm{E}}).$$

*Proof.* For each $U_k$, let $\eta_k$ be an RDF with the property $\Phi\eta_k = U_k$. Applying Proposition 4.2 to sketches $U^\pi$ and $U_k$, we obtain $\|U_{k+1} - U^\pi\|_\infty \leq \gamma^c\|U_k - U^\pi\|_\infty + \gamma^c\varepsilon_{\mathrm{R}} + \varepsilon_{\mathrm{B}} + \varepsilon_{\mathrm{E}}$. Taking a limsup on both sides over $k$ and rearranging yields the result. □

## 4.1 Concrete example

The analysis presented above is abstract; it provides a generic template for conducting error propagation analysis to show that Sketch-DP converges to a neighbourhood of the true values, and moreover illustrates the dependence of this error on the "richness" of the sketch, and accuracy of the Bellman coefficients. To apply this abstract result to a concrete algorithm, we are required to establish the three error bounds that appear in the statement of Proposition 4.2. The result below shows how this can lead to a concrete result for a novel class of sketches; in particular, proving that computed mean embeddings under these features become arbitrarily accurate as the number of features increases.

**Proposition 4.4.** Consider a sketch $\phi$ whose coordinates are feature functions of the form $\phi_i(z) = \mathbb{1}\{z_1 \leq z < z_{i+1}\}$ ($i = 1, \ldots, m-1$), and $\phi_m(z) = \mathbb{1}\{z_1 \leq z \leq z_{m+1}\}$, where $z_1, \ldots, z_{m+1}$ is an equally-spaced grid over $[G_{\min}, G_{\max}]$, with $G_{\min} = \min\mathcal{R}/(1-\gamma)$, $G_{\max} = \max\mathcal{R}/(1-\gamma)$. Let $\mathcal{T}_\phi^\pi$ be the corresponding Sketch-DP operator given by solving Equation (5) with $\mu = \mathrm{Unif}([G_{\min}, G_{\max}])$, and define a sequence $(U_k)_{k=0}^\infty$ by taking $U_0(x)$ to be the sketch of some initial distribution in $\mathscr{P}([G_{\min}, G_{\max}])$, and $U_{k+1} = \mathcal{T}_\phi^\pi U_k$ for all $k \geq 0$. Let $U^\pi \in (\mathbb{R}^m)^{\mathcal{X}}$ be the mean embeddings of the true return distributions. Finally, let $\|\cdot\|$ be the norm on $\mathbb{R}^m$ defined by $\|u\| = \frac{G_{\max}-G_{\min}}{m}\sum_{i=1}^m |u_i|$. Then we have

$$\limsup_{k\to\infty} \|U_k - U^\pi\|_\infty \leq \frac{(G_{\max}-G_{\min})(3+2\gamma)}{(1-\gamma)m}.$$

## 5 EXPERIMENTS

We first conduct a broad empirical investigation into the effects of three key factors in Equation (8): the base feature $\kappa$, the number of features $m$, and the slope $s$, using three tabular MRPs (details in Appendix C.1, extended results in Appendix D.1). As in the example in Section 3.2, we compare the mean embeddings estimated by Sketch-DP with ground-truth mean embeddings, reporting their squared $L^2$ distance (**mean embedding squared error**), and also compare the **Cramér distance** $\max_{x \in \mathcal{X}} \ell_2^2(\hat{\eta}(x), \eta^\pi(x))$ (see e.g. Rowland et al. (2018)) between the distribution $\hat{\eta}(x)$ imputed from the Sketch-DP estimate, and the ground-truth return distribution $\eta^\pi(x)$. To aid interpretation of the Cramér distance results, we also report the Cramér distance between the ground truth $\eta^\pi(x)$ and two baselines. First, the Dirac delta $\delta_{V^\pi(X)}$ at the mean return; we expect Sketch-DP to outperform this naïve baseline by better capturing properties of the return distribution beyond the mean. Second, the return distribution estimate computed by categorical DP (Rowland et al., 2018; Bellemare et al., 2023), a well-understood approach to distrbutional RL based on categorical distributions.

The results for sweeps over feature count $m$ and slope $s$ are shown in Figure 4. By sweeping over $m$, we see that the estimated mean embedding goes towards the ground-truth as we use more features. Further, the Cramér distance also decreases as $m$ increases, suggesting that the distribution represented also approaches the ground-truth. To highlight differences between various Sketch-DP algorithms, we also compute the **excess Cramér**: the Cramér distance $\max_{x \in \mathcal{X}} \ell_2^2(\hat{\eta}(x), \eta^\pi(x))$ as above, minus the corresponding distance between the categorical projection of $\eta^\pi$ (c.f. the red stems in Figure 2) and $\eta^\pi$ itself. All distributional methods perform well on these tasks, and significantly outperform the Dirac estimator in stochastic environments; we note that all methods have tunable hyperparameters (bin locations for CDRL, feature parameters for Sketch-DP), which should inform the interpretation of these results, and in particular direct comparison between methods. The results of the sweep on the slope parameter $s$ show different trends depending on the metric. For smoother $\phi$, generally we can obtain smaller error on the mean embeddings, but the Cramér distances are only small for intermediate range of slope values. This result is expected: when the features are too smooth or too sharp, there exists regions within the return range where the feature values do not vary meaningfully. This results in a more lossy encoding of the return distribution, indicating the importance of tuning the slope parameter of the translation family (Equation (8)).

### 5.1 DEEP REINFORCEMENT LEARNING

We also verify that the Bellman sketch framework is robust enough to apply in combination with deep reinforcement learning. To do so, we aim to learn neural-network predictions $U_\theta(x, a)$ of sketch values for each state-action pair $(x, a)$ in the environment. To be able to define greedy policy improvements based on estimated sketch values, we precompute *value-readout coefficients* $\beta \in \mathbb{R}^m$ by solving $\arg\min_\beta \mathbb{E}_{G \sim \mu}[(G - \langle \beta, \phi(G) \rangle)^2]$, so that we can predict expected returns from the sketch value as $\langle \beta, U_\theta(x, a) \rangle$. This allows us to define a greedy policy, and therefore a Q-learning-style update rule, which given an observed transition $(x, a, r, x')$, first computes $a' =$

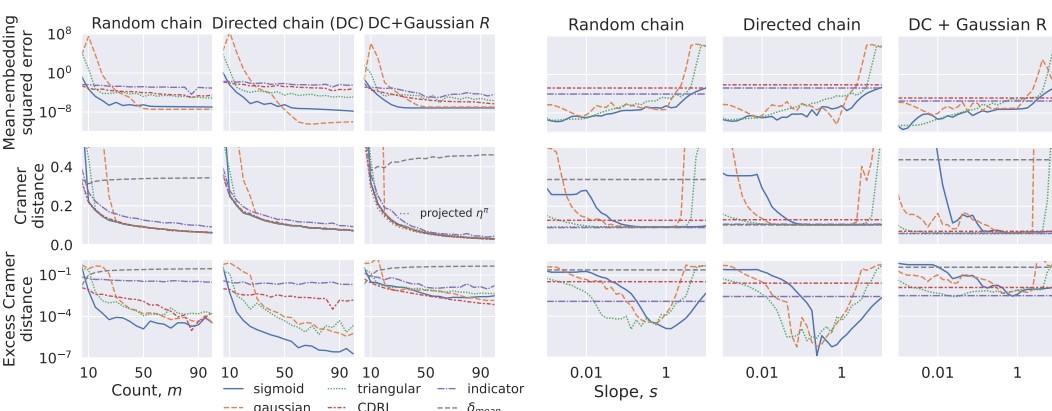

Figure 4: Results of running Algorithm 1 on tabular environments.

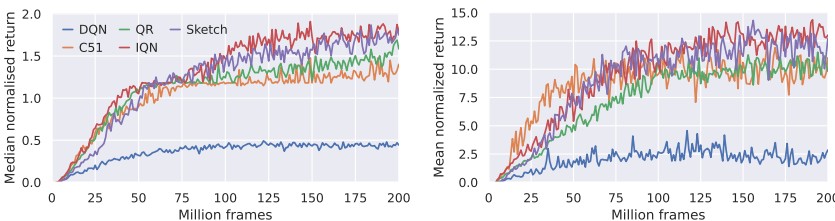

Figure 5: Median (left) and mean (right) human-normalised scores on the Atari 57 suite.

$\arg\max_{\tilde{a}} \langle \beta, U_{\bar{\theta}}(x', \tilde{a}) \rangle$, and then the gradient: $\nabla_\theta \|U_\theta(x, a) - B_r U_{\bar{\theta}}(x', a')\|_2^2$, where $\bar{\theta}$ are the target network parameters. In our experiments, we parametrise $U_\theta$ according to the architecture of QR-DQN (Dabney et al., 2018b), so that the $m$ outputs of the network predict the values of the $m$ coordinates of the corresponding sketch value. We use the sigmoid function as the base feature $\kappa$. Full experimental details for replication are in Appendix C.2; further results are in Appendix D.2.

Figure 5 shows the mean and median human-normalised performance on the Atari suite of environments (Bellemare et al., 2013) across 200M training frames, and includes comparisons against DQN (Mnih et al., 2015), as well as the distributional agents C51 (Bellemare et al., 2017), QR-DQN (Dabney et al., 2018b), and IQN (Dabney et al., 2018a). Sketch-DQN attains higher performance on both metrics relative to the comparator agents C51 and QR-DQN, and approaches the performance of IQN, which uses a more complex prediction network to make non-parametric predictions of the quantile function of the return. These results indicate that the sketch framework can be reliably applied to deep RL, and we believe further investigation of the combination of this framework and deep RL agents is a promising direction for future work.

## 6    RELATED WORK

Typical approaches to distributional RL focus on learning approximate distributions directly (see, e.g., Bellemare et al. (2017); Dabney et al. (2018b); Yang et al. (2019); Nguyen-Tang et al. (2021); Wu et al. (2023)). Much prior work has considered statistical functionals of the random return, at varying levels of generality with regard to the underlying Markov decision process model. See for example Mandl (1971); Farahmand (2019) for work on characteristic functions, Chung & Sobel (1987) for the Laplace transform, Tamar et al. (2013; 2016) for variance, and Sobel (1982) for higher moments. Our use of finite-dimensional mean embeddings is inspired by distributed distributional codes (DDCs) from theoretical neuroscience (Sahani & Dayan, 2003; Vértes & Sahani, 2018; Wenliang & Sahani, 2019), which can be regarded as neural activities encoding return distributions. DDCs were previously used to model transition dynamics and successor features in partially observable MDPs (Vértes & Sahani, 2019). Tano et al. (2020) consider applying non-linearities to rewards themselves, rather than the return, and learning with a variety of discount factors, to encode the distribution of rewards at each timestep. The sketches in this paper are in fact mean embeddings into finite-dimensional reproducing kernel Hilbert spaces (RKHSs; the kernel corresponding to the feature function $\phi$ is $K(z, z') = \langle \phi(z), \phi(z') \rangle$). Kernel mean embeddings have previously been used in RL for representing state-transition distributions (Grünewälder et al., 2012; Boots et al., 2013; Lever et al., 2016; Chowdhury & Oliveira, 2023), and maximum mean discrepancies in RKHSs (Gretton et al., 2012) have been used to define losses in distributional RL by Nguyen-Tang et al. (2021).

## 7    CONCLUSION

We have proposed a framework for distributional reinforcement learning based on Bellman updates that take place entirely within the sketch domain. This has yielded new dynamic programming and temporal-difference learning algorithms as well as novel error propagation analysis, and we have provided further empirical analysis in the context of a suite of tabular MRPs, as well as demonstrating that the approach can be successfully applied at scale as a variant of the DQN architecture. We expect that there will be benefits from further exploration of algorithmic possibilities opened up by this framework, as well as potential consequences for value representations in the nervous system.

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

# Distributional Bellman Operators over Mean Embeddings: Supplementary Material

## A    PROOFS

**Proposition 4.1. (Regression error to Bellman approximation.)** Let $\|\cdot\|$ be a norm on $\mathbb{R}^m$. Then for any return-distribution function $\eta \in \mathscr{P}([G_{\min}, G_{\max}])^{\mathcal{X}}$, we have

$$\max_{x \in \mathcal{X}} \|\Phi(\mathcal{T}^\pi \eta)(x) - (\mathcal{T}_\phi^\pi \Phi \eta)(x)\| \leq \sup_{g \in [G_{\min}, G_{\max}]} \max_{r \in \mathcal{R}} \|\phi(r + \gamma g) - B_r \phi(g)\| . \tag{9}$$

*Proof.* Let $(G(x) : x \in \mathcal{X})$ be an instantiation of $\eta$ (Bellemare et al., 2023); that is, a collection of random variables such that for each $x \in \mathcal{X}$, we have $G(x) \sim \eta(x)$. First, note that the distribution $(\mathcal{T}^\pi \eta)(x)$ is exactly the distribution of $R + \gamma G(X')$ (when the transition begins at $x$ and is generated by $\pi$). So we have

$$\Phi(\mathcal{T}^\pi \eta)(x) = \mathbb{E}_{Z \sim (\mathcal{T}^\pi \eta)(x)}[\phi(Z)] = \mathbb{E}_x^\pi[\phi(R + \gamma G(X'))] .$$

It then follows that:

$$\begin{aligned}
\max_{x \in \mathcal{X}} \|\Phi(\mathcal{T}^\pi \eta)(x) - (\mathcal{T}_\phi^\pi \Phi \eta)(x)\| &= \max_{x \in \mathcal{X}} \left\| \mathbb{E}_x^\pi\Big[\phi(R + \gamma G(X'))\Big] - \mathbb{E}_x^\pi\Big[B_R \mathbb{E}[\phi(G(X'))|X']\Big] \right\| \\
&= \max_{x \in \mathcal{X}} \left\| \mathbb{E}_x^\pi\Big[\phi(R + \gamma G(X')) - B_R \phi(G(X'))\Big] \right\| \\
&\leq \max_{x \in \mathcal{X}} \mathbb{E}_x^\pi\Big[\big\|\phi(R + \gamma G(X')) - B_R \phi(G(X'))\big\|\Big] \\
&\leq \max_{x \in \mathcal{X}} \max_{g \in [G_{\min}, G_{\max}]} \max_{r \in \mathcal{R}} \|\phi(r + \gamma g) - B_r \phi(g)\| ,
\end{aligned}$$

as required.    □

**Proposition 4.2. (Error propagation.)** Consider a norm $\|\cdot\|$ on $\mathbb{R}^m$, and let $\|\cdot\|_\infty$ be the norm on $(\mathbb{R}^m)^{\mathcal{X}}$ defined by $\|U\|_\infty = \max_{x \in \mathcal{X}} \|U(x)\|$. Let $d$ be a metric on return-distribution functions (RDFs) such that $\mathcal{T}^\pi$ is a $\gamma^c$-contraction with respect to $d$. Suppose the following bounds hold.

- (Bellman approximation bound.) For any $\eta \in \mathscr{P}([G_{\min}, G_{\max}])^{\mathcal{X}}$,

$$\max_{x \in \mathcal{X}} \|\Phi(\mathcal{T}^\pi \eta)(x) - (\mathcal{T}_\phi^\pi \Phi \eta)(x)\| \leq \varepsilon_{\mathsf{B}} .$$

- (Reconstruction error bound.) For any $\eta, \bar\eta \in \mathscr{P}([G_{\min}, G_{\max}])^{\mathcal{X}}$ with sketches $U, \bar{U}$, we have $d(\eta, \bar\eta) \leq \|U - \bar{U}\|_\infty + \varepsilon_{\mathsf{R}}$.

- (Embedding error bound.) For any $\eta', \bar\eta' \in \mathscr{P}([G_{\min}, G_{\max}])^{\mathcal{X}}$ with sketches $U', \bar{U}'$, we have $\|U' - \bar{U}'\|_\infty \leq d(\eta', \bar\eta') + \varepsilon_{\mathsf{E}}$.

Then for any two return-distribution functions $\eta, \bar\eta \in \mathscr{P}([G_{\min}, G_{\max}])^{\mathcal{X}}$ with sketches $U, \bar{U}$ satisfying $\|U - \bar{U}\| \leq \delta$, we have

$$\|\Phi\mathcal{T}^\pi \eta - \mathcal{T}_\phi^\pi \bar{U}\|_\infty \leq \gamma^c(\delta + \varepsilon_{\mathsf{R}}) + \varepsilon_{\mathsf{R}} + \varepsilon_{\mathsf{E}} .$$

*Proof.* We follow the illustration laid out in Figure 3:

$$\begin{aligned}
\|\mathcal{T}_\phi^\pi U - \Phi\mathcal{T}^\pi \bar\eta\|_\infty &\overset{(a)}{\leq} \|\mathcal{T}_\phi^\pi U - \Phi\mathcal{T}^\pi \eta\|_\infty + \|\Phi\mathcal{T}^\pi \eta - \Phi\mathcal{T}^\pi \bar\eta\|_\infty \\
&\overset{(b)}{\leq} \varepsilon_{\mathsf{B}} + \|\Phi\mathcal{T}^\pi \eta - \Phi\mathcal{T}^\pi \bar\eta\|_\infty \\
&\overset{(c)}{\leq} \varepsilon_{\mathsf{B}} + d(\mathcal{T}^\pi \eta, \mathcal{T}^\pi \bar\eta) + \varepsilon_{\mathsf{E}} \\
&\overset{(d)}{\leq} \varepsilon_{\mathsf{B}} + \gamma^c d(\eta, \bar\eta) + \varepsilon_{\mathsf{E}} \\
&\overset{(e)}{\leq} \varepsilon_{\mathsf{B}} + \gamma^c(\delta + \varepsilon_{\mathsf{R}}) + \varepsilon_{\mathsf{E}} ,
\end{aligned}$$

as required, where (a) follows from the triangle inequality, (b) follows from the Bellman approximation bound, (c) follows from the embedding error bound, (d) follows from $\gamma^c$-contractivity of $\mathcal{T}^\pi$ with respect to $d$, and (e) follows from the reconstruction error bound. □

**Proposition 4.4.** Consider a sketch $\phi$ whose coordinates are feature functions of the form $\phi_i(z) = \mathbb{1}\{z_1 \leq z < z_{i+1}\}$ ($i = 1, \ldots, m-1$), and $\phi_m(z) = \mathbb{1}\{z_1 \leq z \leq z_{m+1}\}$, where $z_1, \ldots, z_{m+1}$ is an equally-spaced grid over $[G_{\min}, G_{\max}]$, with $G_{\min} = \min \mathcal{R}/(1-\gamma)$, $G_{\max} = \max \mathcal{R}/(1-\gamma)$. Let $\mathcal{T}_\phi^\pi$ be the corresponding Sketch-DP operator given by solving Equation (5) with $\mu = \text{Unif}([G_{\min}, G_{\max}])$, and define a sequence $(U_k)_{k=0}^\infty$ by taking $U_0(x)$ to be the sketch of some initial distribution in $\mathscr{P}([G_{\min}, G_{\max}])$, and $U_{k+1} = \mathcal{T}_\phi^\pi U_k$ for all $k \geq 0$. Let $U^\pi \in (\mathbb{R}^m)^{\mathcal{X}}$ be the mean embeddings of the true return distributions. Finally, let $\|\cdot\|$ be the norm on $\mathbb{R}^m$ defined by $\|u\| = \frac{G_{\max} - G_{\min}}{m} \sum_{i=1}^m |u_i|$. Then we have

$$\limsup_{k \to \infty} \|U_k - U^\pi\|_\infty \leq \frac{(G_{\max} - G_{\min})(3 + 2\gamma)}{(1-\gamma)m} .$$

*Proof.* We begin by obtaining reconstruction and embedding error bounds for this sketch. We introduce the shorthand $\Delta = (G_{\max} - G_{\min})/m$. To obtain a reconstruction error bound, for any distribution $\nu \in \mathscr{P}([z_1, z_{m+1}])$, define $\Pi\nu$ to be the distribution obtained by mapping each point of support $z$ of $\nu$ to the greatest $z_i$ less than or equal to $z$. Mathematically, if we define $f(z) = \max\{z_i : z_i \leq z\}$, then $\Pi\nu = f_\#\nu$, i.e. $\Pi\nu$ is the pushforward of $\nu$ through $f$. We then have $w_1(\nu, \Pi\nu) \leq \Delta$ for all $\nu$ supported on $[z_1, z_m]$, where $w_1$ is the 1-Wasserstein distance, since $f$ transports mass by at most $\Delta$. Introducing another distribution $\nu'$ and the projection $\Pi\nu'$, we note that $w_1(\Pi\nu, \Pi\nu') = \|\Phi\nu - \Phi\nu'\|$. Combining these observations with the triangle inequality yields

$$w_1(\nu, \nu') \leq w_1(\nu, \Pi\nu) + \|\Phi\nu - \Phi\nu'\| + w_1(\nu', \Pi\nu') \leq \|\Phi\nu - \Phi\nu'\| + 2\Delta ,$$

which gives the required form of reconstruction bound, with $\varepsilon_R = 2\Delta$, for the supremum-Wasserstein distance $\overline{w}_1(\eta, \eta') = \max_{x \in \mathcal{X}} w_1(\eta(x), \eta'(x))$ defined over RDFs $\eta, \eta' \in \mathscr{P}(\mathbb{R})^{\mathcal{X}}$. We can also essentially reverse the argument to get

$$\|\Phi\nu - \Phi\nu'\| = w_1(\Pi\nu, \Pi\nu') \leq w_1(\Pi\nu, \nu) + w_1(\nu, \nu') + w_1(\nu', \Pi\nu') \leq w_1(\nu, \nu') + 2\Delta$$

which gives the required form of the embedding error bound, with $\varepsilon_E = 2\Delta$.

Additionally, we can analyse the worst-case regression error $\|\phi(r + \gamma g) - B_r\phi(g)\|$ to get a bound on the Bellman approximation $\varepsilon_B$, by Proposition 4.1. Observe that $\phi(g)$ is constant for $g \in [z_i, z_{i+1})$, and equal to

$$(\underbrace{1, \ldots, 1}_{i \text{ times}}, 0, \ldots, 0)^\top .$$

The minimum regression error in

$$\mathbb{E}_{G \sim \text{Unif}([z_1, z_m])}[\|\phi(r + \gamma G) - B_r\phi(G)\|] \tag{10}$$

is therefore obtained by setting the $i^{\text{th}}$ column of $B_r$ so that

$$B_r\phi(z_i) = \mathbb{E}_{G \sim \text{Unif}([z_i, z_{i+1}))}[\phi(r + \gamma G)] ;$$

note the support of the distribution in the line above. Since $r + \gamma G$ in this expectation varies over an interval of width $\gamma\Delta$, the integrand $\phi(r + \gamma G)$ takes on at most two distinct values. It then follows that we can bound the minimum regression error in Equation (10) by $\Delta$, and hence we can take $\varepsilon_B = \Delta$.

Finally, we observe that $\mathcal{T}^\pi$ maps $\mathscr{P}([G_{\min}, G_{\max}])$ to itself, since for any $g \in [G_{\min}, G_{\max}]$ and any $r \in \mathcal{R}$, we have by construction of $G_{\min}, G_{\max}$ that $r + \gamma g \in [G_{\min}, G_{\max}]$. In addition, we have $\{\Phi\nu : \nu \in \mathscr{P}([G_{\min}, G_{\max}]\} = \{u \in \mathbb{R}^m : 0 \leq u_1 \leq \cdots \leq u_{m-1} \leq u_m = 1\}$, and by the inspection of the columns of $B_r$ above, it follows that $\mathcal{T}_\phi^\pi$ maps $\{\Phi\nu : \nu \in \mathscr{P}([G_{\min}, G_{\max}]\}^{\mathcal{X}}$ to

itself. Therefore the conclusion of Proposition 4.3 holds, and we obtain

$$
\begin{aligned}
\limsup_{k \to \infty} \|U_k - U^\pi\|_\infty &\leq \frac{1}{1-\gamma}(\gamma \varepsilon_R + \varepsilon_B + \varepsilon_E) \\
&\leq \frac{1}{1-\gamma}(\gamma 2\Delta + \Delta + 2\Delta) \\
&= \frac{\Delta(3 + 2\gamma)}{1-\gamma} \\
&= \frac{(G_{max} - G_{min})(3 + 2\gamma)}{(1-\gamma)m}
\end{aligned}
$$

as required. □

## B FURTHER DETAILS AND EXTENSIONS

In this section, we collect further details on a number of topics raised in the main paper.

### B.1 CATEGORICAL IMPUTATION

In the tabular experiments in Sections 3.2 and 5, we include comparisons of distributions imputed from the learned mean embeddings, to provide an interpretable comparison between the different Sketch-DP methods studied. Here, we provide a detailed description of the imputation method.

For a given feature map $\phi$, and a learned sketch value $u$, the goal is to define an imputation strategy $\iota : \mathbb{R}^m \to \mathscr{P}(\mathbb{R})$ (Rowland et al., 2019; Bellemare et al., 2023); that is, a function with the property $\mathbb{E}_{Z \sim \iota(u)}[\phi(Z)] \approx u$, so that $\iota$ serves as an approximate pseudo-inverse to the mean embedding. Here, we follow the approach of Song et al. (2008), and impute probability distributions supported on a finite support set $\{z_1, \ldots, z_n\}$. We define $\iota(s)$ implicitly through the following (convex) quadratic program

$$
\underset{p \in \Delta_n}{\arg\min} \left\| \sum_{i=1}^n p_i \phi(z_i) - s \right\|_2^2.
$$

Note that the left-hand term inside is the expectation of $\phi(Z)$ with $Z \sim \sum_{i=1}^n p_i \delta_{z_i}$, and so the objective is simply aiming to minimise the squared error between the learned sketch value and the sketch value from this discrete distribution. Since this quadratic program is convex, it is solvable efficiently; in our implementations, we use SciPy's MINIMIZE algorithm (Virtanen et al., 2020).

### B.2 CHOICES OF FEATURE FUNCTION

In the main paper, we note that any set of features spanning the degree-$m$ polynomials is Bellman closed, as described by Rowland et al. (2019), and hence exact dynamic programming is possible with this feature set, as shown by Sobel (1982). However, in preliminary experiments we found these features difficult to learn with temporal-difference methods beyond small values of $m$, due to the widely varying magnitude of moments as $m$ grows, making learning rate selection problematic in stochastic environments; further details are provided in Appendix D.3.

In this paper, we tested the following functions as the base feature $\kappa$ in the translation family Equation (8):

- Sigmoid: $\kappa(x) = \frac{1}{1+\exp(-x)}$;
- Gaussian: $\kappa(x) = \exp(-x^2/2)$;
- Hyperbolic tangent: $\kappa(x) = \tanh(x)$
- Morbid: $\kappa(x) = \exp(-x^2/2)\cos(x)$
- Triangular: $\kappa(x) = 1 - |x|$ for $|x| \leq 1$, zero otherwise.
- Parabolic: $\kappa(x) = 1 - x^2$ for $|x| \leq 1$, zero otherwise.

- Quartic: $\kappa(x) = (1 - x^2)^2$ for $|x| \leq 1$, zero otherwise.

In Appendix D.1, we provide further experimental results for a wide variety of feature maps from this family. In addition, we also consider the indicator feature used in Proposition 4.4.

We found that the anchor points must be chosen carefully so that the features produce variations within the return range. This can be done by choosing the range of the anchors to be slightly wider than and estimated return range, and setting the slope so that there does not exist a region of the return range that produce no change in the feature functions. In the tabular experiments, we set the extremum anchor points to be $\hat{G}_{\min} - a\hat{L}$ and $\hat{G}_{\max} + a\hat{L}$, where $\hat{L} = \hat{G}_{\min} - \hat{G}_{\min}$ is the estimated return range, and $a$ is a small positive value around 0.4 casually chosen and not tuned. The return limits $\hat{G}_{\min}$ and $\hat{G}_{\max}$ are estimated by the sample minimum and maximum from samples collected by first-visit Monte Carlo; see Appendix C.1.

The slope parameter should depend on the feature and the return range, and we applied the following intuition. Most of the base feature $\kappa$ have an "non-trivial support" that produces the most variations in the function value. For example, the "non-trivial" support for the sigmoidal and Gaussian $\kappa$ can be chosen as $[-2, 2]$; and for base features that are nonzero only in $[-1, 1]$, this support is $[-1, 1]$. We define the width $w$ of a base function as the length of the non-trivial support. Crudely, the feature with slope $s$ has width $w/s$, as sharper features tend to have shorter non-trivial support. In addition, for the set of features to cover return range uniformly, we set each adjacent feature functions to overlap by 50%. Finally, we want the union of the non-trivial supports of 10 (arbitrary chosen) such overlapping features to equal the return range. We must then have $0.5 \times 10w/s = \hat{G}_{\min} - \hat{G}_{\max}$. This is the *default slope* for each feature and each environment with known return range. As such, the sigmoidal and Gaussian base features have default slope equal to $s = 20/(\hat{G}_{\min} - \hat{G}_{\max})$.

### B.3 CHOICES OF REGRESSION DISTRIBUTION $\mu$

In the main paper, we note that the one-dimensional integrals defining the matrices $C$ and $C_r$ which in turn define the Bellman coefficients $B_r$ can be computed in a variety of ways, depending on the choice of $\mu$ and feature map $\phi$. In our experiments, we take $\nu$ to be a finitely-supported grid in the range $[\hat{G}_{\min} - b\hat{L}, \hat{G}_{\max} + b\hat{L}]$, where $b$ is casually chosen to be around 0.2. The support of $\nu$ is thus slightly wider than the estimated return range and slightly narrower than the anchor range described in Section B.2. The intuition for using a wider anchor range is that we need the features to cover the return distribution (and $\nu$) with the non-trivial support of the features. We validate this intuition in an additional experiment in Appendix D.1. With this $\nu$, Equation (5) is a standard regression problem, and $C$ and $C_r$ can be computed with standard linear-algebraic operations.

Another possibility, particularly if one wishes to use $\mu$ which is not finitely supported, is to use numerical integration to compute these integrals. Additionally, in certain settings the integrals may be computed analytically. For example, with Gaussian $\phi_i(x) = \exp(-s^2(x - z_i)^2/2)$ and Gaussian $\mu$, or $\mu$ as Lebesgue measure (in which case, technically, we modify the expectation in Equation (5) into an integral against an unnormalised measure), $C$ and $C_r$ can be computed analytically. In the case of $\mu$ as Lebesgue measure, we have

$$C_{ij} = \sqrt{\frac{\pi}{2s}} \exp\left(-\frac{s}{2}(z_i - z_j)^2\right) \text{ and } (C_r)_{ij} = \sqrt{\frac{\pi}{s(1+\gamma^2)}} \exp\left(-\frac{s(r + \gamma z_i - \gamma z_j)^2}{1 + \gamma^2}\right).$$

### B.4 KNOWLEDGE OF REWARDS

In distributional approaches to dynamic programming, it is necessary to know all aspects of the environment's transition structure and reward structure in advance, including the set $\mathcal{R}$ required for precomputing the Bellman coefficients. However, in temporal-difference learning, this is a non-trivial assumption. In many environments, this information is available in advance (in the Atari suite with standard reward clipping post-processing (Mnih et al., 2015), rewards are known to lie in $\{-1, 0, 1\}$, for example). When this information is not available, one may modify Algorithm 1 to instead compute Bellman coefficients for observed rewards *just-in-time*; that is, when these rewards are encountered in a transition. This makes the algorithm more broadly applicable, but clearly incurs a significant cost of computing Bellman coefficients for rewards for which these coefficients are not already cached. As Remark 3.2, one possibility in this setting is to learn an approximator

$H : \mathbb{R} \to \mathbb{R}^{m \times m}$ that maps from rewards to Bellman coefficients, and use the predictions of the approximator as proxies for the true Bellman coefficients to reduce the need to solve for the Bellman coefficients every time a new reward is encountered.

## B.5 MATHEMATICAL PROPERTIES OF BELLMAN COEFFICIENTS

The Bellman coefficients $B_r$ play a crucial role in our Bellman sketch framework. Here, we present various properties of $B_r$ in a worked example, derived from both a sigmoid and a Gaussian base feature in Figure 6. For each base feature, we choose 20 evenly spaced anchors in $[-8, 8]$, and find $B_r$ for $r = 1$ and $\gamma = 0.8$, and $\mu$ uniformly supported on a dense grid of 10,000 evenly spaced points in $[-5, 5]$. We apply a small $L^2$ regulariser with weight $10^{-6}$ in the regression problem.

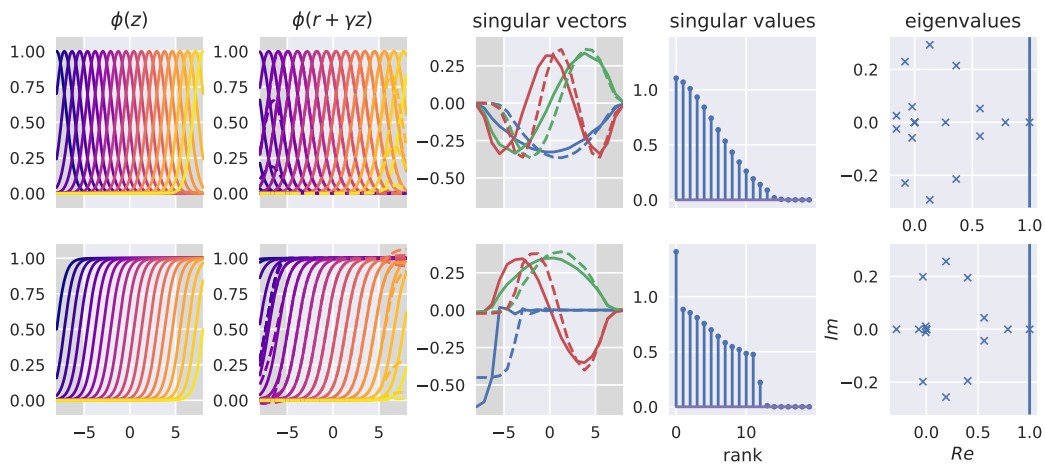

Figure 6: In-depth analysis of Bellman coefficients in the setting described in Section B.5. In the third column, solid curves are the most significant input/right singular vectors, and dashed lines with matching colours are the corresponding output/left singular vectors.

First, we assess how accurate approximation in Equation (6) is when $B_r$ is found via the regression problem in Equation (5). In the left two columns of Figure 6, we show the feature functions $\phi(z)$ and $\phi(r + \gamma z)$ in the first two columns. In the second column, we also show $B_r \phi(z)$ in dashed lines evaluated on $[-8, 8]$, wider than the grid over which we minimised the error. The error is tiny and virtually invisible within the interval $[-5, 5]$, but is larger outside. Quantitatively, the maximum absolute difference between $\phi(r + \gamma z)$ and $B_r \phi(z)$ over the dense grid in $[-5, 5]$ is less than 0.002 for both base features. By Proposition 4.1, we expect a small error in a single step of dynamical programming.

Had $B_r$ been a contraction, we would have been able to prove contraction for Algorithm 1. However, we show empirically that $B_r$ is not in general a contraction in $L^2$ norm, but the dynamics from repeated multiplication of $B_r$ may converge to a stable fixed point. First, we performed a singular value decomposition of the $B_r$ for the two base features. We see that the singular vectors in Figure 6 (third column) are similar to harmonic functions. Importantly, in the largest singular values (operator norms) in Figure 6 (fourth column) are greater than 1, suggesting that a single application of $B_r$ may expand the input. Further, we show the eigenvalues of $B_r$ in the fifth column of Figure 6. Interestingly, all eigenvalues have real parts less or very close to 1.0 suggesting that there exists fixed points in the dynamics induced by $B_r$. As such, the Bellman coefficients $B_r$ exhibit transient dynamics (typical for non-normal matrices) but is stable after repeated applications to an initial vector. Further studies into these dynamical properties are important for future work. Given the important role of non-normal dynamics hypothesised to be present in the nervous system (Hennequin et al., 2012; Bondanelli & Ostojic, 2020), these observations allude to the possibility that the Bellman sketch framework could contribute to a biological implementation of distributional RL.

### B.6 COMPUTATIONAL PROPERTIES OF BELLMAN COEFFICIENTS

Under many choices of feature maps $\phi$, the matrix $B_r$ has structure that may be exploited computationally. We provide sketches of several cases of interest. For "binning features", even for overlapping bins, $B_r$ is a very narrow band matrix, and hence is sparse, leading to linear-time matrix-vector product computation. This remains approximately true for other forms of localised features, such as low-bandwidth Gaussians and related bump-like functions, and in particular applying truncation to near-zero coefficient in the Bellman coefficients in such cases will also lead to sparse matrices.

**Bellman coefficients as least-squares coefficients.** The closed-form solution for the Bellman coefficients $B_r$ in Equation (7) can be derived by viewing the optimisation problem in Equation (5) as a vector-valued linear regression problem, and using the usual expression for the optimal prediction coefficients. The derivation is the same in content to the usual derivation of least-squares coefficients, which we provide below for completeness, to illustrate how it is obtained in our case. We begin by differentiating the (quadratic) objective in Equation (7) with respect to $B$, and setting the resulting expression equal to the zero vector, to obtain

$$-2\mathbb{E}_{G\sim\mu}[\phi(r+\gamma G)\phi(G)^\top] + 2\mathbb{E}_{G\sim\mu}[B_r\phi(G)\phi(G)^\top] = 0\,.$$

Rearranging, we obtain

$$B_r\mathbb{E}_{G\sim\mu}[\phi(G)\phi(G)^\top] = \mathbb{E}_{G\sim\mu}[\phi(r+\gamma G)\phi(G)^\top]\,.$$

Finally, under the assumption of invertibility of $\mathbb{E}_{G\sim\mu}[\phi(G)\phi(G)^\top]$, we obtain the expression for the Bellman coefficients in Equation (7):

$$B_r = \mathbb{E}_{G\sim\mu}[\phi(r+\gamma G)\phi(G)^\top]\mathbb{E}_{G\sim\mu}[\phi(G)\phi(G)^\top]^{-1}\,.$$

**Online computation of Bellman coefficients in the case of unknown rewards.** In settings where the set of possible rewards $\mathcal{R}$ is not known in advance, is infinite, or is too large to cache Bellman coefficients for all possible rewards $r \in \mathcal{R}$, we may exploit the structure of the Bellman coefficients described above to speed up the computation of the coefficients online. Rather than solving the regression problem from scratch, an alternative is to cache the matrix $\mathbb{E}_{G\sim\mu}[\phi(G)\phi(G)^\top]^{-1}$ above, and construct the matrix $\mathbb{E}_{G\sim\mu}[\phi(r+\gamma G)\phi(G)^\top]$ as required, upon observing a new reward $r$. This reduces the marginal cost of computing the Bellman coefficients $B_r$ to a matrix-matrix product.

### B.7 COMPARISON WITH OTHER APPROACHES TO DISTRIBUTIONAL RL

In this section, we provide additional comparisons against existing approaches to distributional RL, including their theoretical analysis. As distributional RL is a quickly evolving field, we focus our comparison on a few main classes of algorithms related to our work, which illustrate some key axes of variation within the field: (i) categorical approaches (Bellemare et al., 2017); (ii) quantile approaches (Dabney et al., 2018a;b; Yang et al., 2019); (iii) approaches related to maximum mean discrepancy (MMD; Gretton et al., 2012), such as Nguyen-Tang et al. (2021); Zhang et al. (2021); Sun et al. (2022); and (iv) sketch-based approaches (Sobel, 1982; Rowland et al., 2019).

**Distribution representation.** Categorical, quantile, and MMD approaches are typically presented as learning approximate return distributions directly. In categorical approaches, the approximate distribution is parametrised as

$$\sum_{i=1}^m p_i\delta_{z_i}\,,$$

with fixed particle locations $(z_i)_{i=1}^m$, and learnable probabilities $(p_i)_{i=1}^m$ for each state-action pair at which the return distribution is to be approximated. In contrast, quantile and MMD approaches learn fixed-weight particle approximations, of the form

$$\sum_{i=1}^m \frac{1}{m}\delta_{z_i}\,,$$

in which the particle locations $(z_i)_{i=1}^m$ are learnable. Work on sketches has instead focused on learning the values of particular statistical functionals of the return, rather than explicitly approximating

return distributions. Rowland et al. (2019) also shows that standard categorical- and quantile-based algorithms can also be viewed through the lens of sketch-based distributional RL. The approach proposed in this paper sits firmly in the camp of sketch-based approaches, without ever representing approximated distributions directly. We highlight generative models of distributions as another prominent class of (non-parametric) representation (see, e.g., Doan et al. 2018; Freirich et al. 2019; Dabney et al. 2018a; Yang et al. 2019; Wu et al. 2023).

**Algorithm types.** Most prior algorithmic contributions to distributional reinforcement learning have focused on sample-based temporal-difference approaches, in which prediction parameters are iteratively and incrementally updated based on the gradient of a sampled approximation to a loss function. These approaches include the original C51 (Bellemare et al., 2017), QR-DQN (Dabney et al., 2018b), MMDRL (Nguyen-Tang et al., 2021), and EDRL (Rowland et al., 2019) algorithms. Dynamic programming algorithms, in which parameters are not updated incrementally via loss gradients, but instead according to the application of an implementable operator, have also been considered (see Rowland et al. (2018; 2023); Wu et al. (2023) for categorical dynamic programming, quantile dynamic programming, and fitted likelihood estimation, respectively). In this paper, our algorithmic contributions include both DP and TD methods.

**Losses and projections.** One of the core axes of variation across distributional RL approaches is the loss used to define updates in incremental algorithms, and to define projections in dynamic programming. Categorical approaches use a projection in Cramér metric (Rowland et al., 2018) to define a target distribution for both dynamic programming and incremental versions of the algorithm; the incremental algorithm updates predictions via the gradient of a KL loss between the current and target distributions. Quantile-based approaches use either the quantile regression loss in incremental settings, or a Wasserstein-1 projection in dynamic programming (Dabney et al., 2018b). The MMDRL approach has been proposed only in the incremental setting, and considers sample-based gradients through an MMD loss, specifically taking the form

$$\mathrm{MMD}_K^2 \left( \sum_{i=1}^m \frac{1}{m} \delta_{z_i(x,a)}, \sum_{i=1}^m \frac{1}{m} \delta_{r+\gamma z_i(x',a')} \right), \tag{11}$$

for some choice of kernel $K$. In contrast, the Sketch-DP and Sketch-TD algorithms introduced in this paper define updates via matrix-vector products with the Bellman coefficients derived in Equation (5). With the connection to RKHS noted in Section 6, one can view the squared norm appearing in Equation (5) as the squared norm in the RKHS generated by the feature functions $\phi$. From this perspective, it is clear that both the Bellman coefficients that feature in Sketch-DP and Sketch-TD, and the gradients of the loss in Equation (11), both relate to the mathematical structure of RKHS.

Contrasting against the Sketch-DP/Sketch-TD approaches described above, earlier approaches to sketched-based distributional RL, both in dynamic programming and incremental forms, have defined losses via imputation strategies, which compute updates by converting sketches into approximate distributions (Rowland et al., 2019; Bellemare et al., 2023). Foreshadowing the remarks on theoretical analysis below, we remark that neither MMDRL nor the earlier sketch-based approach described above have been analysed for convergence, while Section 4 in this paper deals with convergence analysis of the approach proposed in this paper.

**Theoretical analysis.** The analysis of both dynamic programming and sample-based incremental algorithms requires analysis of the interaction between distributional updates, and approximations made to the distributions due to the choice of representation mentioned above.

Convergence analysis of dynamic programming algorithms has been obtained for several classes of distributional algorithms; see Rowland et al. (2018) for the case of categorical dynamic programming, Dabney et al. (2018b) for the case of a quantile dynamic programming algorithm, and Bellemare et al. (2023); Rowland et al. (2023) for later generalisations of this work. This analysis typically centres around (i) proving contractivity of the distributional Bellman operator $\mathcal{T}^\pi$ with respect to some metric $\mathrm{d}$, and proving non-contractivity of the specific distributional projection used by the dynamic programming algorithm under this same metric. Notably, the metric $\mathrm{d}$ used in the *analysis* need not be the same as any metrics used in *defining* the algorithm; this is the case for quantile dynamic programming, for which Wasserstein-1 distance is used to define the algorithm, while Wasserstein-$\infty$ distance is used to analyse the algorithm (Dabney et al., 2018b; Bellemare et al., 2023; Rowland et al., 2023). Our proof technique, in particular, makes use of contraction of

the distributional Bellman operator in Wasserstein distances, though such distances do not feature in the definition of the Sketch-DP/TD algorithms.

In general, there has been less work on the convergence analysis of sample-based incremental algorithms. Rowland et al. (2023) recently showed convergence of quantile temporal-difference learning, though the question of convergence for many other sample-based incremental distributional reinforcement learning algorithms is currently open. In particular, we note as a point of comparison to Nguyen-Tang et al. (2021), that although this work provides a contraction analysis of $\mathcal{T}^\pi$ (Theorem 2), and MMD approximation bounds for fixed target distributions (Theorem 3 and Proposition 2), these do not constitute a proof of convergence of the incremental algorithm described therein. The analysis of incremental algorithms is generally more mathematically involved than in the dynamic programming case, principally owing to the fact that rather than analysing the iterated application of a fixed operator, one needs to analyse the continuous dynamical system associated with incremental updates.

## C    EXPERIMENTAL DETAILS

In this section, we provide additional details on the experimental results reported in the main paper.

### C.1    TABULAR ENVIRONMENTS

We describe the setup in the main paper. In Appendix D.1, we show extended results of more features and more environments.

**Environments.** In the main paper, we reported results on the on the following environments,

- **Random chain:** Ten states $\{x_1, x_2, \ldots, x_{10}\}$ are arranged in a chain. There is equal probability of transitioning to either neighbour at each state, and state $x_{10}$ has a deterministic reward of +1;

$$\text{terminal} \leftarrow x_1 \longleftrightarrow x_2 \longleftrightarrow x_3 \longleftrightarrow \cdots \longleftrightarrow x_{10} \rightarrow \text{terminal}.$$

- **Directed chain (DC):** Five states are arranged in a directed chain, but the agent can only move along the arrow deterministically until termination. A deterministic reward of +1 is given at state $x_5$;

$$x_1 \longrightarrow x_2 \longrightarrow x_3 \longrightarrow \cdots \longrightarrow x_5 \rightarrow \text{terminal}.$$

- **DC with Gaussian reward:** A variant of the directed chain above, with the only difference that $x_5$ has a Gaussian reward with mean 1 and unit variance.

The discount factor is $\gamma = 0.9$. These environments cover stochastic and deterministic rewards and state transitions, giving a range of different types of return distributions.

**Feature functions.** We use features of translation family Equation (8), with $\kappa$ chosen from a subset of base features described in equation (B.2). We also include the indicator features used in Proposition 4.4. For the sweep over slope, we set the slope $s$ to be the default slope (described in Appendix B.2) multiplied by a scaling factor, and sweep over this factor. from 0.001 to 10.0. This is done primary because the return range varies a lot across different environments, and the default slope is adjusted to the return range. The results serve as justification for the heuristics on choosing the default slope.

**Ground-truth distribution.** We approximate the ground-truth mean embeddings and the ground-truth return distributions by collecting a large number of return samples from the MRPs. To do so, we use first-visit Monte Carlo with a sufficiently long horizon (after the first visit to each state) to ensure that the samples are unbiased and has bounded error caused by truncating the rollout to a finite horizon. For environments with deterministic rewards, truncating the horizon at $L$ steps induces maximum truncation error $|r|_{\max}\gamma^L/(1-\gamma)$, where $|r|_{\max}$ is the maximum reward magnitude. We bound this error at $10^{-4}$, giving $L > 110$, so we set the horizon after the first visit to 110. For environments with Gaussian rewards, we set the horizon to 200. We initialise the rollout at each state in the environment, and for initial each state this is repeated $10^5$ times. This gives us at least $10^5$ samples each state.

**Sketch DP under conditional independence.** Many RL environments, including the tabular environments tested in this paper, have the property that $R \perp\!\!\!\perp X'|X$ for the trajectory $X, A, R, X'$, so the Sketch-DP update Equation (Sketch-DP) simplifies to

$$U(x) \leftarrow \mathbb{E}_x^\pi[B_R]\mathbb{E}_x^\pi[U(X')] = \mathbb{E}_x^\pi[B_R] \sum_{x' \in \mathcal{X}} P(x'|x)U(x').$$

This means we need to evaluate the expected Bellman coefficient $\mathbb{E}_x^\pi[B_R]$. This is trivial for deterministic rewards. For stochastic rewards with known distributions, we approximate the expectation via numerical integration. We run all DP methods for 200 iterations.

**Jittered imputation support.** The support on which we impute the distribution are the anchors of the features. Some tabular environments have states with deterministic returns that directly align with the feature anchors, which interferes in unintuitive ways with the finite support on which we impute distributions, producing non-monotonic trends in the results. To avoid this unnecessary complication, we jitter the support before imputing the distribution: for points in the support, we add noise uniformly distributed over $[-\Delta/2, \Delta/2]$, where $\Delta$ is the distance between consecutive support points. Likewise, we project ground truth distribution using the same jittered support to. In Figure 4, we report the average of the metrics computed from 100 independent jitters. Note that since the imputation (from mean embedding) and projection (from ground-truth) share the same support for each of the 100 jitters, the average Cramér distance between the projected and the ground-truth still lower-bounds the average Cramér distance between the imputed distribution and the ground-truth.

## C.2  DEEP REINFORCEMENT LEARNING IMPLEMENTATION DETAILS

In this section, we provide further details on the deep reinforcement learning experiments described in the main paper, in particular describing hyperparameters and relevant sweeps.

**Environment.** We used the exact same Atari suite environment for benchmarking QR-DQN (Bellemare et al., 2013; Dabney et al., 2018b). In all experiments, we run three random seeds per environment.

**Feature map $\phi$.** The results in Figure 5 uses the sigmoid base feature $\kappa(x) = 1/(e^{-x} + 1)$ with slope $s = 5$, and the anchors to be 401 (tuned from 101, 201 and 401) evenly spaced points between $-12$ and 12. These values are loosely motivated by the range used in C51 (Bellemare et al., 2017). We did not use the heuristic in Appendix B.2 to choose the slope parameter, instead we set this to 10 by tuning from $\{1, 2, \ldots, 12\}$. Larger slope values typically resulted in Bellman coefficients with a worst-case regression error $\max_{r \in \mathcal{R}} \max_{g \in \text{supp}(\mu)} \|\phi(r + \gamma g) - B_r \phi(g)\|$ greater than 0.01. In these cases, we regard the regression error as too large, and did not perform agent training with these hyperparameter settings. In addition, $\phi$ is appended with a constant feature of ones, which we found to be very crucial for a good performance; as noted in the main paper, this ensures that the sketch operator is truly affine, not linear. We also tried a several other feature functions, including the Gaussian, the hyperbolic tangent and the (Gaussian) error function, and found that the sigmoid reliably performed the best.

**Solving the regression problems.** To compute the Bellman coefficients as well as the value readout weights $\beta$ described in Section 5.1, we solve the corresponding regression problem with $\mu$ set to be 100,000 points evenly spaced between $-10$ and 10. We also add a $L^2$ regulariser with strength set to $10^{-9}$ to avoid numerical issues, which is tuned from $\{10^{-15}, 10^{-12}, 10^{-9}, 10^{-6}, 10^{-3}\}$.

**Neural network.** We implement Sketch-DQN based on the QR-DQN architecture, using exactly the same convolutional torso and fully-connected layers to estimate the mean embeddings. The differences are:

- We add a sigmoid or tanh nonlinearity, depending on the base feature output range to the final layer. This helps bound the predicted mean embedding and improved the results.
- The network only predicts the non-constant dimensions of the mean embedding, and the constant feature is appended as a hard-coded value.
- We use the pre-computed mean readout coefficients $\beta$ to predict state-action values for state-action pairs in the Q-learning objective, and at current states to determine the greedy policy.

**Training.** We use the exact same training procedure as QR-DQN (Dabney et al., 2018b). Notably, the learning rate, exploration schedule, buffer design are all the same. We tried a small hyperparameter sweep on the learning rate, and found the default learning rate $0.00005$ to be optimal for performance taken at 200 million frames.

**Evaluation.** The returns are normalised against random and human performance, as reported by Mnih et al. (2015). We use the mean and median over all games and three random seeds for each game.

**Baseline methods.** We also tuned the number of atoms of the approximating distributions in the baseline methods. In particular, we found that C-51 performed the best compared to using more atoms; IQN did best when using 51 quantiles; and QR-DQN did best using 201 quantiles. Increasing the number of atoms in these methods lead to worse performance. We report the results of these best variants of the corresponding baseline methods in Figure 5.

## D  FURTHER EXPERIMENTS

In this section, we collect further experimental results to complement those reported in the main paper.

### D.1  EXTENDED TABULAR RESULTS

We tested Sketch-DP using the following additional MRPs. If unspecified, the default reward distribution for each state is a Dirac delta at 0, and the transition probabilities from a state to its child states are equal.

- Tree: State 1 transitions to states 2 and 3; state 3 transitions to states 4 and 5. State 2 has mean reward 5; state 4 has mean reward -10, and state 5 has mean reward 10. All leaf states are terminal.
- Loopy tree: Same as Tree, but with a connection from state 2 back to state 1.
- Cycle: Five states arranged into a cycle, with only a single state having mean reward 1.
- Rowland '23: The environment in Example 6.5 of Rowland et al. (2023).
- S&B '18: The environment in Example 6.4 of Sutton & Barto (2018).
- Loopy fork: The environment shown in Figure 2(A).

All environments have discount factor $\gamma = 0.9$. For each environment, except Rowland '23 and S&B'18, the reward distributions for the non-zero-reward states are either Dirac deltas at the specified mean, or Gaussian with specified mean and unit standard deviation.

The results, extending those in Figure 4, are illustrated in Figure 7. The results are in general consistent with the main Figure 4. In particular, the Cramér distances decay as the number of features increases, and can be closer to the corresponding projected ground-truths than the CDRL baseline.

In Appendix B.2, we suggested that the range of the anchors should be wider than the range of the uniform grid $\mu$ on which we measure the regression loss. We validate this intuition by performing another experiment, sweeping the ratio of the width of the anchor range relative to the width of the grid $\mu$, fixing the mid points between these two ranges the same. Here, we use $m = 50$ features for each base feature function and apply the default slope described in Appendix B.2. The results in Figure 8 shows that a slightly wider anchor range produces reliably small Cramér distances for almost all base features. When the anchor range decreases from 1 to 0, there is a much sharper increase in the Cramér distance, because the support on which we impute the distribution is too narrow and can miss substantial probability mass outside the support. On the other hand, when the anchor range increases from 1, the Cramér distance also increases because the support points get further away from each other, lowering the resolution of the imputed distribution. Since the grid is chosen to be slightly wider than the true return range, we see that the smallest Cramér distances can be attained at anchor range ratio slightly less than 1, but this does not hold universally (see, e.g., random chain and cycle results). Choosing this ratio to be slightly greater than 1, as suggested in Appendices B.2 and B.3, is more reliable at the cost of a small increase in distributional mismatch.

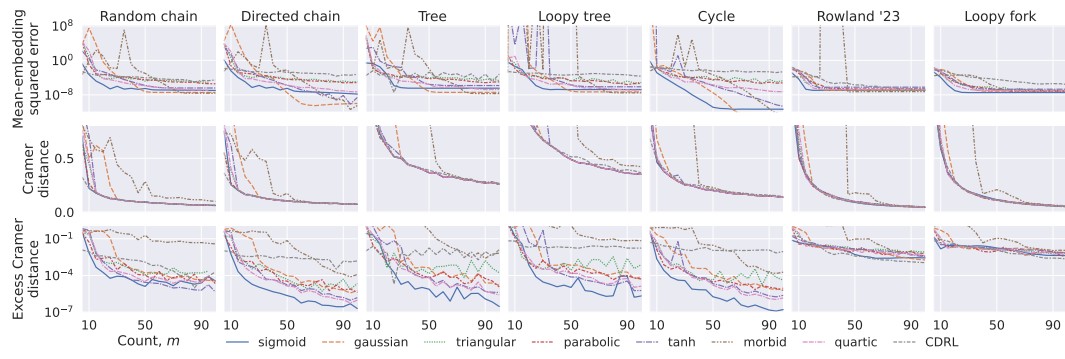

(a) Environments with deterministic rewards, sweeping over feature count $m$.

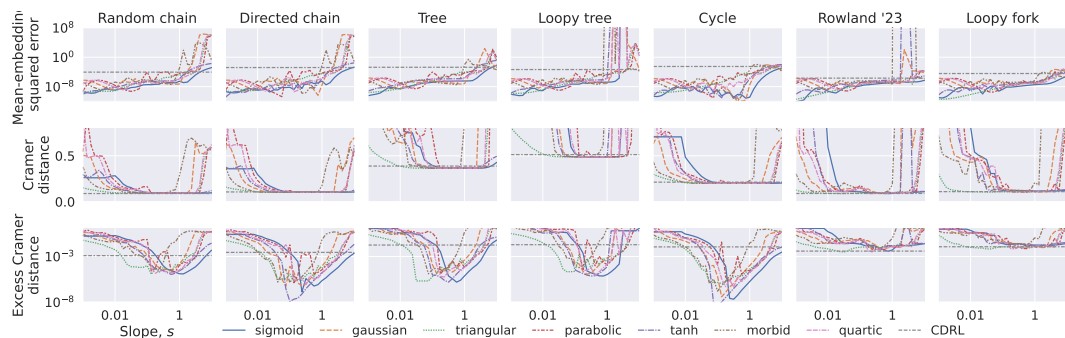

(b) Environments with deterministic rewards, sweeping over slope $s$.

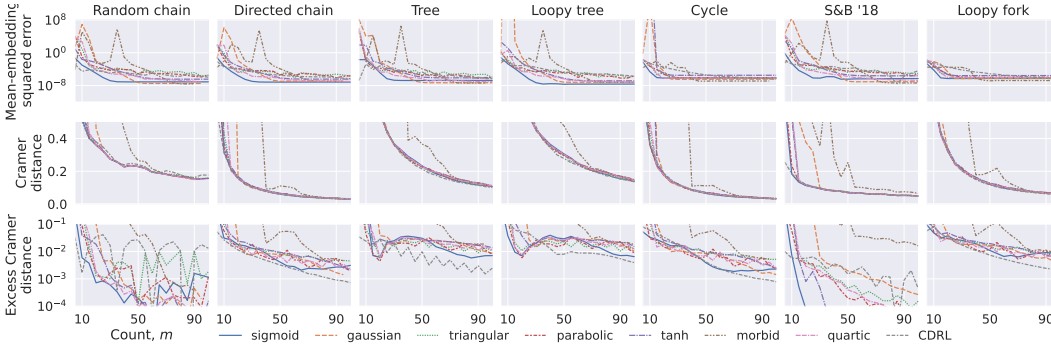

(c) Environments with Gaussian rewards, sweeping over feature count $m$.

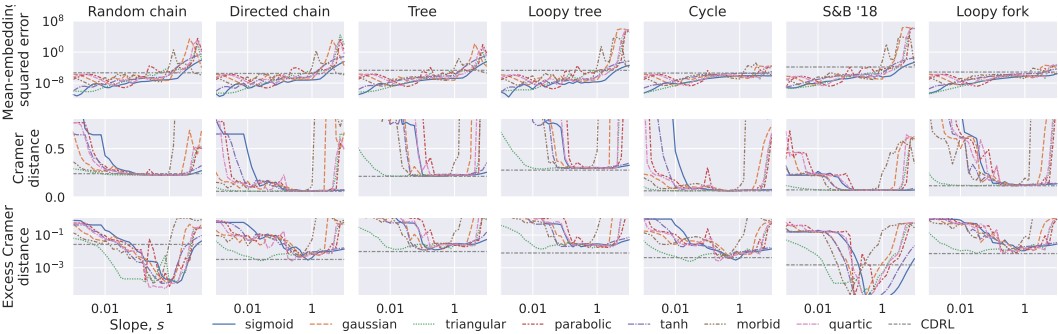

(d) Environments with Gaussian rewards, sweeping over slope $s$.

Figure 7: Additional tabular results extending Figure 4.

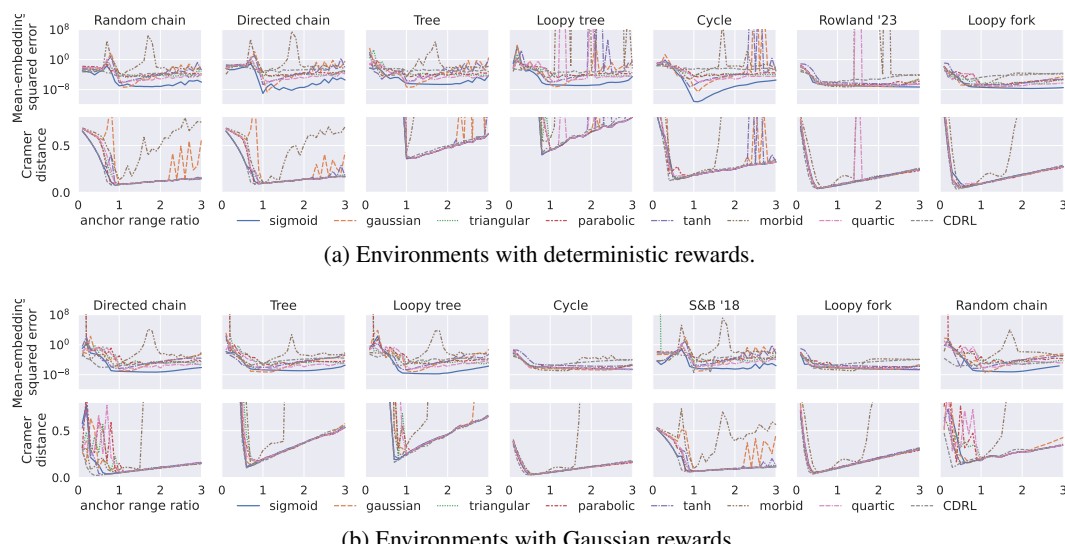

(a) Environments with deterministic rewards.

(b) Environments with Gaussian rewards.

Figure 8: Additional tabular results from sweeping over the range of the anchors relative to the width of the uniform grid given by the support of $\mu$.

### D.2 EXTENDED RESULTS ON ATARI SUITE

We show in Figure 9 the advantage of the Sketch-DQN method to other baselines. On one hand, we see that Sketch-DQN surpasses DQN on almost all games. Compared to IQN and QR-DQN, Sketch-DQN is consistently better on CRAZY CLIMBER, SPACE INVADERS, RIVER RAID, ROAD RUNNERS and VIDEO PINBALL, while worse on ASSAULT, ASTERIX, DOUBLE DUNK, KRULL, PHOENIX, and STAR GUNNER.

To show how sensitive are the results depending on the feature parameters, we ran the full Atari suite using a few sigmoidal and Gaussian features, and show the results in Figure 10. Overall, we see that the choice on feature parameters is important; in particular, the sigmoidal feature outperforms Gaussian features. For the sigmoidal feature, the performance improved from using 101 to 201 features. On the contrary, for Gaussian features, increasing the feature count does not produce much change.

### D.3 TEMPORAL-DIFFERENCE LEARNING WITH POLYNOMIAL FEATURES

As noted in the main text, the sketch corresponding to the polynomial feature function $\phi(g) = (1, g, g^2, \ldots g^m)$ is Bellman closed, and the Bellman coefficients $B_r$ obtain zero regression error in Equation (5). In addition, there has been much prior work on dynamic programming (Sobel, 1982) for moments of the return, and temporal-difference learning specifically in the case of the first two moments (Tamar et al., 2013; 2016). However, such polynomial feature functions are difficult to use as the basis of learning high-dimensional feature embeddings. This stems from several factors, including that the typical scales of the coordinates of the feature function often vary over many orders of magnitude, making tuning of learning rates difficult, as well as the fact that polynomial features are *non-local*, making it more difficult to decode distributional information via an imputation strategy.

To quantify these informal ideas, we ran an experiment comparing Sketch-TD updates for a 50-dimensional mean embedding based on the translation family (Equation (8)) with a sigmoid base feature $\kappa$, as well as polynomial features with $m = 5$, and $m = 50$. The sigmoid features are chosen according to the intuitions in Appendix B.2. We ran 100,000 synchronous TD updates on mean embedding estimates initialised at $\phi(0)$ for all state. Each run uses a fixed learning rate chosen from $10^{-6}$ to 1.

In Figure 11, we plot the Cramér distance of imputed distributions from ground-truth after running TD for each of these three methods, on a variety of the environments described in Section D.1. In

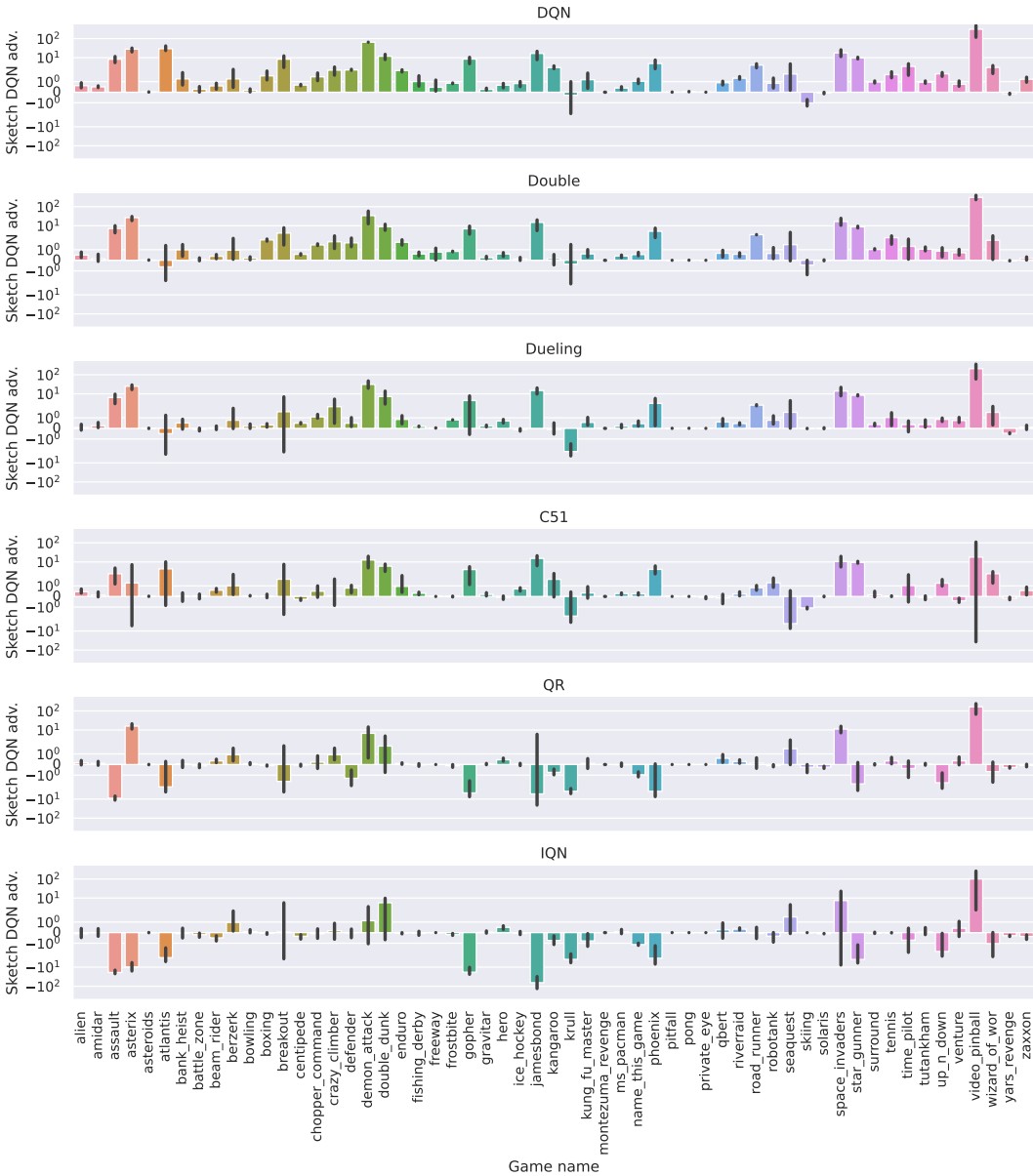

Figure 9: Advantage of Sketch-DQN, measured as Sketch-DQN's normalised return minus the returns of other baseline methods. Positive values means Sketch-DQN is better. We show the mean and standard error over 3 seeds for each game.

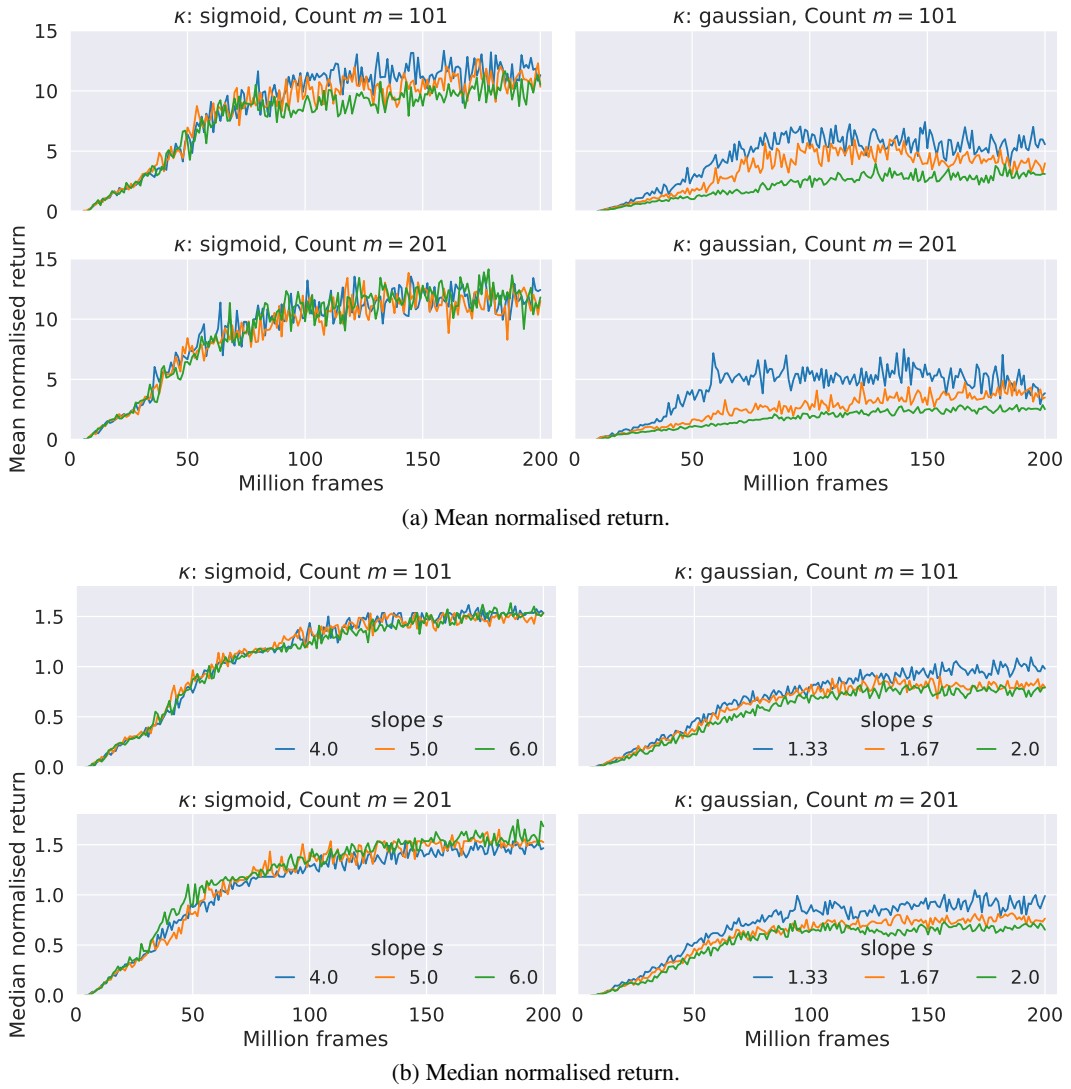

Figure 10: Results on Atari suite for different feature parameters.

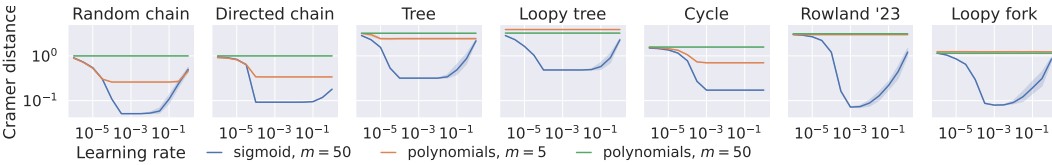

Figure 11: Results comparing sketch-TD with sigmoidal and polynomial features.

all environments, there is a similar pattern. For the Sketch-TD algorithm based on sigmoid non-linearities, there is a reasonably wide basin of good learning rates, with performance degrading as the learning rate becomes too small or too large. On several environments this pattern is reflected also in the performance of the degree-5 polynomial embedding, though the minimal Cramér error is generally significantly worse than that of the sigmoid embedding. This supports our earlier observations; this mean embedding captures relatively coarse information about the return distribution, and in addition different feature components have different magnitudes, meaning that a constant learning rate cannot perform well. The mean embedding with degree-50 polynomials generally performs very badly on all environments, as the components of the embedding are at such different magnitudes that no appropriate learning rate exists.

### D.4 COMPARISON WITH STATISTICAL FUNCTIONAL DYNAMIC PROGRAMMING

The Sketch-DP methods developed in this paper were motivated in Section 2 with the aim of having distributional dynamic programming algorithms that operate directly on sketch values, without the need for the computationally intensive imputation strategies associated with SFDP algorithms. In this section, we empirically compare Sketch-DP and SFDP methods, to quantitatively measure the extent to which this has been achieved. We give details below of the Sketch-DP and SFDP algorithms we compare, and provide comparisons of per-update wallclock time to assess computational efficiency, and distribution reconstruction error to assess accuracy.

**Sketch-DP.** We consider the Sketch-DP algorithm based on sigmoid features as described in Section 3.1, and implemented as described in Section 5 and Appendix C.1.

**SFDP.** We consider the SFDP algorithm for learning expectile values as described by Bellemare et al. (2023, Section 8.6). We use SciPy's default `minimize` implementation (Virtanen et al., 2020) to solve the imputation strategy optimisation problem given in Equation (8.15) in Bellemare et al. (2023). For a given sketch dimension $m$, we use expectiles at linearly spaced levels $\tau_i = (2i - 1)/(2m)$ for $i = 1, \ldots, m$.

**Results.** These two algorithms, for varying numbers of feature/expectiles $m$, were run on a selection of deterministic-reward environments as described in Appendix D.1. In Figure 12, we plot the Cramér distance and the excess Cramér distance of reconstructed distributions to ground truth, as described in Section 5 and plotted in Figure 4. In addition, we also plot two wallclock times in each case: the average time it takes to run one iteration of the dynamic programming procedure, and the time it takes to setup the Bellman operator, which includes solving for $B_r$ for Sketch-DP. As predicted, the run time is significantly higher for the SFDP algorithm, due to its use of imputation strategies. The approximation errors measured by Cramér distances are also smaller for Sketch-DP, particularly as the number of features/expectiles is increased. Considering the per-update wallclock times in the third row of the figure, there is consistently a speed up of at least 100x associated with the Sketch-DP algorithm relative to SFDP. This is due to the fact that the Sketch-DP update consists of simple linear-algebraic operations, while the SFDP update includes calls to an imputation strategy, which must solve an optimisation problem. The one-off computation of the Bellman coefficients takes around 0.1–4 seconds, depending on the number of features $m$, which is only at most a couple of SFDP iterations, and hence a small fraction of the total run time of the SFDP algorithm.

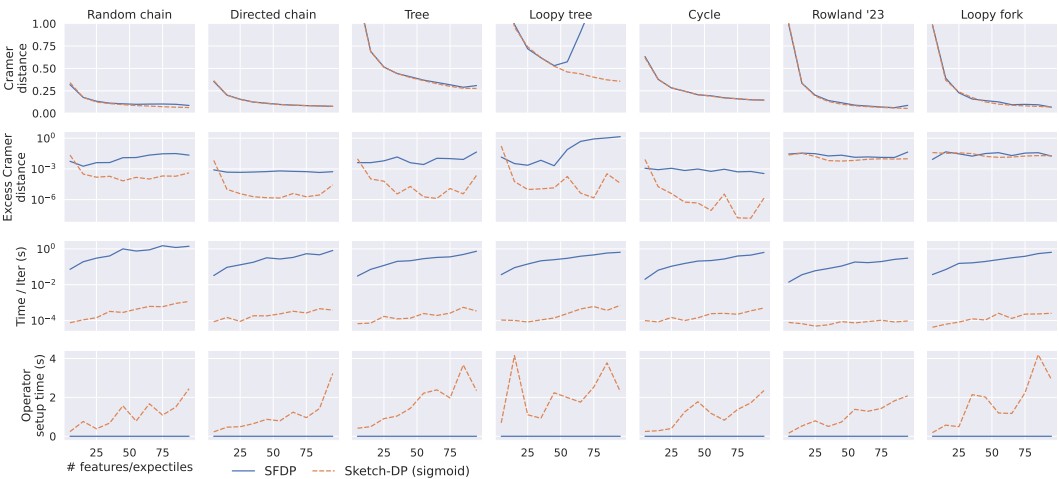

Figure 12: Results comparing Cramér distances as in Figure 4 (first two rows), and wallclock runtimes for each DP iteration (third row) and for setting the corresponding Bellman operator (bottom row), for Sketch-DP and SFDP algorithms, varying the numbers of features/expectiles $m$.

