# OpenReview forum: "Distributional Bellman Operators over Mean Embeddings"
_ICLR.cc/2024/Conference — Submitted to ICLR 2024_

### Official Review · Reviewer_7TXx · 2023-10-27

**Soundness:** 3 good
**Presentation:** 2 fair
**Contribution:** 1 poor
**Rating:** 3
**Confidence:** 5

**Summary:**

This paper studies the mean embedding sketches, one kind of statistical functional, in the context of distributional RL. They first show the sketch framework, where the Bellman operator is applied on the sketch instead of the original value function in classical RL. Further, they also provide the convergence guarantee under the sketch framework. Experiments are conducted on both tabular MDP and Atari games.

**Strengths:**

* The writing is clear. The paper is well-organized and easy to follow.

* Experiments are extensive by considering both MDP and all Atari games with 200M frames.

**Weaknesses:**

* **Limited Methodology contribution and novelty**. Firstly, the sketch is just one special statistical functionals equipped with a specified kernel function, which is typically less commonly used than quantiles or samples. Working on algorithms simply based on a new concept may not contribute to the real development of this research field. More importantly, feature mapping has already been implicitly considered in MMDDRL paper [1], where MMD naturally induces a feature map. Notably. [1] also shows that not all kernels will lead to a convergent distributional Bellman operator, about which this paper has not fully discussed. From the perspective of new algorithms, Sketch-DP or TD is not new to me and even straightforward. Its concrete version, e.g., in Proposition 4.4 is very similar to the categorical representation. The choice of different feature maps also has a strong correlation with the MMD equipped with different kernels. Based on my knowledge of RKHS, different kernel naturally induces a RHKS with a specific feature map. Therefore, I do not think the sketch framework has sufficient methodological contribution compared with existing works.

* **Insufficient theoretical analysis**. The biggest issue in the theoretical part is the pre-specification of a metric $d$ such that it is $\gamma^c$ contractive, which is a strong assumption. With a metric that can already guarantee the contraction, the approximation error is easy to show. Further, as mentioned in the first weakness, not all feature maps can guarantee the contraction. For example, [1] gave some counterexamples for the convergence when we use Gaussian kernels. Therefore, the current paper lacks a crucial theoretical part about what kinds of feature maps can guarantee the contraction and then bound the errors. Without this crucial part, I personally think the current theoretical results are insufficient.

* **Insignificant empirical improvements**. The viewpoint of limited methodological contribution is further demonstrated by the insignificant empirical improvements in Figure 5. Although I truly agree on the conclusion that the sketch framework slightly improves C51 and QRDQN, it performs worse than IQN. Hence, I only view the sketch paper as a feature map-based algorithm, which has already been implicitly investigated before, rather than a novel one with significant performance. The insignificant improvement also let me rethink what is the real motivation of the sketch framework.


[1] Thanh Nguyen-Tang, Sunil Gupta, and Svetha Venkatesh. Distributional reinforcement learning via
moment matching. (AAAI 2021)

**Questions:**

Please refer to the Weakness.

---

> ### Author Response · Authors · 2023-11-16
>
> We thank the reviewer for their time in reviewing the paper.
>
> We respectfully disagree with the reviewer's assessment, and in particular we would like to clarify fundamental differences between our work and that of Nguyen-Tang et al. (2021), and clarify the validity and novelty of our theoretical results. We hope that our response below makes these points clear, and that the reviewer will consider revising their score; we would be very happy to engage in further discussion on any of these points if there is remaining uncertainty.
>
> **Methodology contribution, and novelty**\
> The reviewer seems to claim that the methodology introduced in the paper is not novel, and is already encompassed by the work of Nguyen-Tang et al. (2021), stating "Sketch-DP or TD is not new to me". We strongly disagree with this, and we believe the works are fundamentally different, for the following reasons:
> * The work of Nguyen-Tang et al. contributes an empirically highly successful approach to distributional temporal-difference learning. The core features of their approach are:
>     * The approach approximates return distributions by learning particle locations, as with QR-DQN.
>     * These particle locations are learnt through temporal-difference learning, by combining an MMD loss with target distributions formed by distributional bootstrapping.
>     * As mentioned above, this approach is highly successful empirically, but the TD algorithm itself is not analyzed theoretically. The authors prove contractivity of the distributional Bellman operator with respect to certain MMDs, but this is typically only one aspect of many required to analyze a distributional RL algorithm, since the distribution approximation and TD dynamics must also be taken into account.
> * In contrast, this work contributes a family of dynamic programming and temporal-difference learning algorithms for learning sketch values directly. Contrasting the characteristics of our work with those of Nguyen-Tang et al.'s work, we note:
>     * Our approach learns sketch values directly, and does not approximate probability distributions through particle representations.
>     * We derive a dynamic programming algorithm for updating sketch values based on the notion of Bellman coefficients, defined by the regression problem in Eqn (5). This results in a DP procedure expressible through straightforward matrix-vector products. We also derive an associated TD procedure through stochastic approximation of the DP procedure.
>     * We provide a novel theoretical analysis of the complete DP procedure described above, based on the expression of the DP procedure via Bellman coefficients.
>
> Since the work of Nguyen-Tang et al. (2021) and our work therefore take quite different approaches, we do not believe that the claim of lack of novelty of our work is supported. If the reviewer believes our method has appeared elsewhere, we would be happy to take into consideration further references; otherwise, we would be grateful to the reviewer if they could update their score accordingly.
>
> **Theoretical analysis**\
> We believe the reviewer has misunderstood the requirements on the metric $d$. Proposition 3.2 requires that the usual distributional Bellman operator $\mathcal{T}^\pi$ is a contraction with respect to $d$. This is not a strong assumption at all, as standard metrics such as supermum-Wasserstein-$p$ metrics and the supermum-Cramer distance satisfy this requirement.
>
> In addition, we believe there is a high degree of novelty in our theoretical analysis. As mentioned above, we give a complete analysis of a distributional RL algorithm, Sketch-DP, in contrast to analyzing the contractivity of the distributional Bellman operator, which is not sufficient in itself to analyze the behavior of practical distributional RL algorithms, due to not incorporating distribution approximation errors into the analysis (as in Nguyen-Tang et al). This analysis goes beyond the typical contractive mapping approach, and we believe this form of analysis will be useful in future theoretical research on distributional RL.

---

> > ### Comment · Reviewer_7TXx · 2023-11-21
> > **Reviewer Response**
> >
> > Thanks for the author's response, however, I am not satisfied with the authors' explanation.
> >
> > ### 1. Methodology and novelty
> > Firstly, the statement of MMDDRL [1] is inaccurate.
> > * MMDDRL uses particles/samples, while QR-DQN uses quantiles. Sample representation is one of the advantages of [1] over pre-defined statistics, e.g., [2].
> > * MMDDRL is still a theoretically heavy paper, and it may not be appropriate to lower their theoretical contribution in the authors' statement. For example, [1] has discussed the distribution approximation error/sample complexity in Theorem 3 and Proposition 2.
> >
> > Despite these points, the largest concern is the overlap between the paper and [1] as both are RKHS-based methods. To induce an RKHS, a kernel is pre-specified with an implicit feature map. [1] uses the mean embedding distance between two distributions in RKHS, and similarly your work explicitly considers the feature mapping. I acknowledge the methodological differences between you and [1], but the mathematical foundation especially the contraction would be very linked. However, unfortunately, I have not found sufficient discussions about this issue except only a brief statement in the discussion part, which is unsatisfactory.
> >
> > For a practical algorithm in Prop 4.4, it is very linked with CDRL, where the range of value functions should be pre-specified. This is also the major deficiency of CDRL compared with QRDQN. However, it seems that your proposed algorithm also suffers from this fundamental issue.
> >
> >
> > ### 2. Theoretical analysis
> > Linked with 1., a natural question is how you analyze the contraction, regardless of DP or TD, in terms of the kernel. [1] already analyzed, showing that the contraction in DP is correlated with the choice kernel. However, I did not find the relevant theoretical results in the paper, while the only thing I found is the $\gamma^c$ contraction assumption. I am aware that this assumption is true for Wasserstein and Cramer distance, but it is not true for general kernel-based algorithms, which are highly linked with the choice of kernel. This is why I point out this critical issue, although I acknowledge the contribution of the following analysis. It is no doubt non-trivial to have such a detailed answer instead of directly imposing an assumption in the paper.
> >
> > ### 3. Insignificant empirical performance.
> >
> > Another major limitation is the empirical performance is insignificant as it only slightly outperforms QR-DQN and is significantly inferior to IQN. Also, MMDDRL is (almost) the state-of-the-art algorithm that significantly outperforms IQN, FQF. Since both [1] and your algorithm are kernel-based or RKHS-based, why bother to use an algorithm among two similar ones that is infererior in performance?
> >
> > In summary, although I acknowledge the contribution of this paper in the methodology (so called sketch) and some new theoretical analysis, I would keep my score for a rejection mainly based on the three fundamental limitations in my mind.
> >
> > [1] Thanh Nguyen-Tang, Sunil Gupta, and Svetha Venkatesh. Distributional reinforcement learning via moment matching. (AAAI 2021)

---

> ### Author Response · Authors · 2023-11-22
> **Official Comment by Authors:  Methodology and Novelty**
>
> Thank you for the additional comments.
>
> The reviewer states that our description of MMDRL above is inaccurate, although we can't see any substantive inaccuracies highlighted in their response. In response to the two bullet points the reviewer raised:
>
> * Both MMDRL and QR-DQN learn approximate distributions of the form $\sum_{i=1}^m \frac{1}{m} \delta_{z_i}$, and in particular learn the particle locations $(z_i)_{i=1}^m$. For QR-DQN specifically, the particle locations are trained to represent (approximate) quantile locations.
>     * On methodology, the review asserts that a sample representation is advantageous over pre-defined statistics. To our knowledge, this isn't a uniform principle; whether each representation is better than another depends on many factors, and on the application. The reference [2] in the reviewer's response is not included; we would be happy to comment further if the reviewer provides this reference.
> * The reviewer is correct that Theorem 3 and Proposition 2 in Nguyen-Tang et al. analyze approximation error of a *fixed* distribution with a set of particles under MMD. However, these results, and the contraction analysis of $\mathcal{T}^\pi$ in Theorem 2 of that paper, do not prove convergence of the algorithm proposed in that paper; based on this, we don't agree with the reviewer's claim of "lower[ing] their theoretical contribution". Please see the response on theoretical analysis for further comments on this.
>
> ***“the largest concern is the overlap between the paper and [1] as both are RKHS-based methods”***\
> We agree with the reviewer that both methods are related to the mathematical concept of RKHS, but we don't see substantial overlap beyond this, with notable variation along the dimensions of particle/sketch approximation, TD/DP algorithms, and theoretical convergence analysis in the case of Sketch-DP, as described in our first response.
>
> We would like to emphasize that we view the work of Nguyen-Tang et al. as an important contribution in the field of distributional RL, and we believe this paper and Nguyen-Tang et al. make complementary, distinct contributions. To make this clearer, **we have added Appendix B.7**, which summarizes these points of comparison to existing approaches, as well as our response on the point of theoretical analysis below. We have also added a comment regarding the use of MMD by Nguyen-Tang et al. in the related work section.
>
> ***"but the mathematical foundation especially the contraction would be very linked".***\
> We agree with the reviewer that both approaches are related to the mathematical concept of RKHS. However, as mentioned above, we don't think there is a mathematical link between the two works beyond this, and as far as we can tell, the reviewer is not suggesting a concrete link in their response. If the reviewer has a concrete link between mathematical foundations in mind, we would be happy to discuss further.
>
> We also believe the mention of "contraction" here is related to a misunderstanding regarding the relevance of showing contractivity of $\mathcal{T}^\pi$ with respect to certain metrics, described in more detail in the theoretical analysis section of the response below.
>
> ***"This is also the major deficiency of CDRL compared with QRDQN. However, it seems that your proposed algorithm also suffers from this fundamental issue."***\
> The reviewer asserts that the specific instance of the sketch algorithm in Prop 4.4 is similar to CDRL. However, CDRL uses triangular features, whereas this instance uses square indicator functions. The analysis of CDRL relies on the projection of measures on the Cramer distance, facilitated by the triangular features, while our analysis is motivated by bounded error analysis based on the Wasserstein distance. Therefore, CDRL and this method differ both practically and theoretically.
>
> We also disagree that selecting distribution support in advance is a fundamental issue. Return distributions are very often bounded in applications of interest, which leads to a natural choice of return range. This idea is used in several modern high-performing RL agents, including MuZero (Schrittwieser et al., 2020), and DreamerV3 (Hafner et al., 2023).
>
> **References**
>
> Schrittwieser, Julian, et al. "Mastering atari, go, chess and shogi by planning with a learned model." Nature 588.7839 (2020): 604-609.
>
> Hafner, Danijar, et al. "Mastering diverse domains through world models." arXiv preprint arXiv:2301.04104 (2023).

---

> ### Author Response · Authors · 2023-11-22
> **Official Comment by Authors: Theoretical Analysis**
>
> We believe there may be a misunderstanding here about what is required in the analysis of a distributional DP algorithm. We hope the explanation below makes clear that the paper contributes a full rigorous analysis for the introduced class of DP algorithms, and we also highlight that complete analysis of dynamic programming procedures in distributional RL, as in this paper, is a substantial theoretical contribution (see e.g. Rowland et al., 2023, Wu et al., 2023, for other examples). We would be very happy to continue the discussion if the reviewer has further questions.
>
> ***"Linked with 1., a natural question is how you analyze the contraction, regardless of DP or TD, in terms of the kernel."***\
> We believe the reviewer is asking about the contraction specifically of the distributional projection operator $\mathcal{T}^\pi$ (without any projection), with respect to the (supremum-)MMD induced by the choice of feature map.
>
> The central point we would like to make is:\
> Establishing contractivity of $\mathcal{T}^\pi$ with respect to this (supremum-)MMD does not prove convergence of the algorithms we propose in the paper. Conversely, our convergence analysis does not make use of such a result.
>
> Thus, contractivity of $\mathcal{T}^\pi$ with respect to the (supremum-)MMD is not in and of itself a relevant theoretical result for analyzing the algorithms proposed in the paper. Therefore, our response to the following point from the reviewer:
>
> *"However, I did not find the relevant theoretical results in the paper, while the only thing I found is the $\gamma^c$ contraction assumption."*
>
> is that it is not a relevant theoretical result, and our convergence analysis studies the DP algorithm without using such a result, with Proposition 4.4 using supremum-Wasserstein distance, for example.
>
> ***"I am aware that this assumption is true for Wasserstein and Cramer distance, but it is not true for general kernel-based algorithms, which are highly linked with the choice of kernel. "***\
> Given the explanation above, we expect this comment is related to the difference between:
> * A metric used in defining a distributional RL algorithm.
> * A metric used to analyze this algorithm.
>
> We believe our discussion above addresses this concern. Our analysis does not require us to use the same metric used in defining the algorithm to analyze the algorithm.
>
> ***"... I acknowledge the contribution of the following analysis. It is no doubt non-trivial to have such a detailed answer instead of directly imposing an assumption in the paper."***\
> Thank you for this comment. We hope that with the explanations given above, there are no remaining theoretical concerns that the reviewer has, but please let us know if not.
>
> **Further details on the role of contraction in distributional RL convergence results**\
> To provide additional context as to why it is not necessary to prove convergence of $\mathcal{T}^\pi$ with respect to a certain metric to establish convergence results for distributional dynamic programming algorithms, we recall some examples from the literature:
>
> * In the case of quantile dynamic programming (see Rowland et al., 2023, and also Dabney et al., 2018 for earlier analysis), the algorithm is defined with a Wasserstein-1 projection, but the convergence analysis makes use of a result showing contractivity of $\mathcal{T}^\pi$ with respect to the supremum-Wasserstein-$\infty$ metric. This contractivity result is then combined with a non-expansion result for the projection mapping component of quantile-dynamic programming to prove convergence.
> * In the case of fitted likelihood estimation (FLE; Wu et al., 2023), one result that contributes to the analysis of the main algorithm in the paper is the contractivity of $\mathcal{T}^\pi$ with respect to a weighted sum of Wasserstein-2 distances at each state, but note that the Wasserstein-2 distance is unrelated to the proposed algorithm itself. This result is then combined with an error propagation analysis to obtain a proof of convergence.
>
> In contrast, the work of Nguyen-Tang et al.: (i) establishes contractivity of $\mathcal{T}^\pi$ with respect to certain supremum-MMD metrics, (ii) analyzes approximation of *fixed* distributions in MMD. However, these results are not used to prove convergence of the algorithm proposed in the paper, and the algorithm proposed in the paper is not analyzed. Analysis of their proposed TD algorithm would require analysis of the dynamical system induced by the incremental particle updates; see Rowland et al. (2023) for an example in the case of quantile TD learning.

---

> ### Author Response · Authors · 2023-11-22
> **Official Comment by Authors: Empirical Performance**
>
> We agree that the performance of Sketch-DQN on the Atari suite is not superior to all existing distributional methods. We would argue that state-of-the-art performance in the Atari suite should not be necessary for acceptance, given that the paper primarily is concerned with contributing novel algorithmic ideas and theoretical analyses in distributional RL. These deep RL experiments show that our theoretically motivated approach is able to achieve good performance in Atari, achieving similar benefits as observed with earlier approaches to distributional RL.
>
> ***"why bother to use an algorithm among two similar ones that is infererior in performance?"***\
> The reviewer questions the value of having a new algorithm that performs better than all but one benchmark. First, we do not agree with the premise that our algorithm is similar to that of Nguyen-Tang et al. beyond superficial relations in RKHS (see above). Second, we do not intend to convince people to commit to our method a priori, but to show that this method works in combination with deep RL in practice and reaches or surpasses established benchmarks, under the same or similar computational budget, tasks and training procedures.
>
> Distributional RL methods are used in a wide variety of applications, and at a wide variety of scales; whether this method will be superior or inferior to another in any given application depends on a wide variety of factors. The point is that we offered a novel family of DRL algorithms, notably with convergence analysis in the DP setting, with promising results, and hence it is worth considering for other applications.

---

### Official Review · Reviewer_RJqf · 2023-10-28

**Soundness:** 3 good
**Presentation:** 2 fair
**Contribution:** 2 fair
**Rating:** 6
**Confidence:** 3

**Summary:**

This paper presents a novel distributional RL framework based on the sketch Bellman operator. The approach is derived from statistical functional dynamic programming and involves constructing different sketches to formulate the Bellman equation. The authors provide a theoretical guarantee to support their proposed method and empirical studies demonstrate that it outperforms some baselines in Atari environments.

**Strengths:**

1) The authors provide detailed experiments, particularly for ablations on different feature functions.

2) The authors provide theoretical guarantee, which adds credibility to their proposed method.

**Weaknesses:**

1) The authors' motivation for using the sketch Bellman operator is to reduce the need for expensive imputation strategies when converting between sketches and distributions. However, the experiment does not verify this claim. It would be helpful if the authors could provide some quantitative results (such as training time) to demonstrate this reduction.

2) The proposed method performs worse than IQN, even though the authors claim that IQN uses a more complex prediction network for non-parametric predictions. It would be beneficial if the authors could provide quantitative results (such as neural network sizes and training time) to explain this difference.

3) It would be useful to include vanilla Statistical Functional Dynamic Programming (Bellemare et al., 2023) as a baseline in some toy examples to compare its performance with the proposed Sketch-DQN.

4) The writings in some sections are not very clear. For example,  in the subsection on 'Computing Bellman coefficients',  the authors directly show the closed form of (5) without any explanations or citations.

5) There are some typos that need to be addressed, such as in Sec 2.2, where it should be $
\left(\left(\mathcal{T}^\pi \iota (U)\right)(x)\right)$ instead of $
\left(\left(\mathcal{T}^\pi \iota U\right)(x)\right)$.

**Questions:**

Please answer the questions mentioned above.

---

> ### Author Response · Authors · 2023-11-16
>
> We thank the reviewer for their time and constructive feedback in reviewing our paper. We respond to specific queries raised by the reviewer below. We are eager to engage in further discussion on any of these points if the reviewer has further queries, and hope that the reviewer will consider re-evaluating their score as a result.
>
> **Comparison with statistical functional dynamic programming** \
> Based on your suggestion, we ran a version of statistical functional dynamic programming (SFDP) using expectiles and the imputation strategy described by Bellemare et al. (2023) for comparison against Sketch-DP. We use SciPy's default minimization algorithm to compute the imputation strategies required by SFDP. We have included the full results in Appendix D.4 in the latest version of the paper. A brief summary is provided below to provide quick answers to the reviewer's points 1 & 3.
>
> * Point 1. Running SFDP with expectiles, and Sketch-DP with sigmoid features in the environments from the paper, we found that the per-update wallclock time for SFDP was consistently greater than 100x the wallclock time for Sketch-DP. This is due to Sketch-DP requiring only matrix-vector product computations, while SFDP requires the solution of an optimization problem in each call it makes to the imputation strategy.
> * Point 3. With the same experimental set-up, we found that the reconstruction errors (measured in Cramer distance) for Sketch-DP and SFDP to be comparable for low numbers of features/expectiles, with some advantage to the Sketch-DP approach being observed for higher numbers of features/expectiles.
>
> Thus, these results confirm that there are significant practical speed-ups that come from the Sketch-DP approach relative to SFDP. We thank the reviewer for suggesting this additional comparison; full results are given in Appendix D.4, please let us know if you have any further questions on these areas.
>
> **IQN training times.** Based on the suggestions of several reviewers, we reported measured wallclock times for Sketch-DQN, QR-DQN, and IQN in the general response above, to support the discussion around the additional complexity associated with IQN in the paper.
>
> **Derivation of Equation (7).** The form for the Bellman coefficients in Eqn (7) follows since Eqn (5) defines a least-squares linear regression problem, which B_r solves. The expression $C_r C^{-1}$ is the analogue of the usual $\hat{\beta} = (X^T X)^{-1} X^T Y$ expression for the least-squares coefficients for the linear model expressed in the form $Y = X \beta + \epsilon$. We thank the reviewer for raising this question, and agree that this derivation can be made clearer; we have added a guide to the derivation in Appendix B.6, and in doing so, corrected a typo in Eqn (7). Please let us know if you have any further queries.
>
> **Typo.** Thank you for noting this, and please let us know if there are any other typos you had in mind for correction.

---

> > ### Comment · Reviewer_RJqf · 2023-11-21
> > **Thanks for your response.**
> >
> > The authors have addressed my main concerns, so I increase my score from 5 to 6.

---

### Official Review · Reviewer_PQKZ · 2023-11-01

**Soundness:** 3 good
**Presentation:** 2 fair
**Contribution:** 2 fair
**Rating:** 5
**Confidence:** 4

**Summary:**

This paper proposes a novel algorithmic framework for distributional reinforcement learning that works purely in the sketch space based on finite-dimensional mean embeddings of return distributions. The authors derive the approximate Bellman equation in the sketch space and propose a new algorithm that can be combined with dynamic programming and temporal-difference learning. The authors provide an asymptotic convergence theory for the proposed method, and examine the representation error of the proposed sketch on a suite of tabular tasks. They also demonstrate that this approach can be straightforwardly combined with deep reinforcement learning, obtaining a new deep RL agent that improves over the QR-DQN baseline on the Arcade Learning Environment.

**Strengths:**

1. The paper introduces a novel framework for distributional reinforcement learning, which is based on learning mean embeddings of return distributions. This approach avoids the need for expensive imputation strategies, which can be computationally expensive and biologically implausible.
2. The authors provide a theoretical analysis of the proposed algorithms, including asymptotic convergence results. This analysis helps to establish the theoretical foundations of the approach and provides insights into its properties.

**Weaknesses:**

1. The proposed method requires a linear approximation for the Bellman update equation and require calculating a Bellman coefficient matrix $B_r$ that can be computationally challenging. This contradicts the motivation to improve computation efficiency and reduce the imputation error by purely operating in the sketch space. It is unclear how the proposed method is superior to previous methods both computationally and statistically.

2. The experimental validation is limited. The experiment on the tabular tasks only shows the approximation error of the proposed sketch method rather than the performance of the overall approach. From the result, the proposed sketch method has a similar Cramer distance compared to the CDRL baseline. While the proposed sketch has lower excess Cramer distance and mean-embedding squared error, it is unclear how these metrics translate into the performance of the proposed approach. For the Deep RL part, the proposed method underperforms IQN and does not significantly outperform QR-DQN, which is the backbone of the proposed method. Does the proposed method have a significant computation advantage?

Minor: CDRL is mentioned but not referred to in the paper.

**Questions:**

1. What is the advantage of the proposed method compared with previous distributional RL methods? Does the proposed method have a provable lower approximation error or computation advantage?

2. How does the proposed method perform on the tabular tasks compared with baselines? Does the proposed method have a computation advantage in the deep RL setting?

---

> ### Author Response · Authors · 2023-11-16
>
> We thank the reviewer for their time and constructive feedback in reviewing our paper. We respond to specific queries raised by the reviewer below. We are eager to engage in further discussion on any of these points if the reviewer has further queries, and hope that the reviewer will consider re-evaluating their score as a result.
>
> **Computation of Bellman coefficients**\
> The computation time of the Bellman coefficients $B_r$ is insignificant, particularly in the setting of deep reinforcement learning, since it is a one-off computation done before running the recursive updates; see Algorithm 1, where $B_r$ is computed before the main loop. For example, it takes only a couple of seconds on a CPU to compute $B_r$ for the case with 100 feature functions, and $n=10^5$ grid points to estimate the expectations in Eqn (7). This is negligible in the Atari experiments, where only rewards of -1, 0 and 1 are encountered, relative to the many forward/backward passes made in training deep networks. We have also added additional experiments comparing the computation cost associated with Sketch-DP with an earlier distributional RL algorithms (SFDP) in the tabular setting in Appendix D.4 (please see the response to Reviewer RJqf for further details), which confirm that Sketch-DP delivers computational improvements relative to SFDP in the tabular setting too.
>
> Additionally, in settings where possible reward values are not available in advance, dealing with unseen reward values $r$ does not require solving the regression problem from scratch. This is because $C^{-1}$ can be saved, and we only need to compute $C_r$ for unseen values of $r$, which is also very fast; we have added a paragraph to Appendix B.6 to describe the details of this approach.
>
> **Comparison against previous methods in combination with deep reinforcement learning**\
> Based on your suggestion, we have reported new results in the general response above, comparing wallclock rates of frame throughput for Sketch-DQN, QR-DQN, and IQN. As described in detail in the general response, Sketch-DQN attains a higher throughput than QR-DQN and IQN due to the straightforward nature of the distributional update derived for sketches in this paper. We thank the reviewer again for this suggestion. We would also like to emphasize that although QR-DQN and Sketch-DQN share the same network architecture, the two agents use completely different losses for updating their predictions.
>
> Advantage over previous methods. We would like to emphasize that some of the core advantages of the sketch framework introduced in this paper are (i) the generality of the framework - each choice of feature functions yields a new algorithm; and (ii) the way in which these methods can be analyzed, as described in Section 4. We expect these contributions to be of use in future distributional RL work, beyond the current paper. Our experimental results also show benefits of Sketch-DP/TD algorithms, but we emphasize that the contributions of the paper extend beyond the empirical performance of the newly introduced algorithms.
>
> **Focus on approximation error in tabular experiments.** We believe the reviewer is highlighting the difference in control and prediction tasks within reinforcement learning; our deep RL experiments are focused on measuring the performance of Sketch-DQN in control, and in our tabular experiments, we focus on understanding the quality of predictions made by the sketch algorithms introduced in the paper. Measuring approximation error is an important aspect of empirical evaluation of the methods; it allows us to understand how well a particular algorithm approximates the return distribution in various settings, and from this perspective we view it as an important complement to the deep RL experimental results. Further, it has been shown that in tabular settings, distributional reinforcement learning may not necessarily offer improved performance for control (Lyle et al., 2019), meaning that in the tabular setting, evaluation of distribution predictions is often more informative as a means of understanding the algorithm's properties.
>
> **Minor points.** Thank you for mentioning the CDRL acronym, we will ensure it is defined and referenced in the final version.
>
> References
>
> Clare Lyle, Pablo Samuel Castro, Marc G. Bellemare, A Comparative Analysis of Expected and Distributional Reinforcement Learning, AAAI Conference on Artificial Intelligence, 2019.

---

### Official Review · Reviewer_ReLT · 2023-11-01

**Soundness:** 4 excellent
**Presentation:** 4 excellent
**Contribution:** 4 excellent
**Rating:** 8
**Confidence:** 3

**Summary:**

This paper discusses a method to do distributional reinforcement learning using mean embedding sketches. The advantage of this method is that the sketches can be updated without ever obtaining the imputed distribution, thereby allowing computations entirely in the sketch domain.

**Strengths:**

This paper satisfies all the criteria of an excellent paper:
1. It is exceptionally clearly written. Reading it and learning from it was a joy. It does a great job at being thorough without being pedantic in its discussion. I really appreciated the concrete discussion in sections 3.2 and 4.1; it dovetails quite nicely with the rest of the paper.
2. It makes a useful contribution to the fun and important problem of distributional reinforcement learning. The idea is simple yet elegant.

**Weaknesses:**

The formulation of feature maps used in the paper (equation (8)) is sort of "pulled out a hat". It would be useful to justify why it made sense to use that formulation as opposed to other possibilities. See also the questions below.

**Questions:**

1. The general sketch using Bellman coefficient $B_r$ will correspond to some mean embedding sketch which is an invertible linear combination of first $m$-moments, right?
2. Why can we write $B_r$ as $C_rC^{-1}$?
3. Where's the generalization to the case when $\mathcal R$ is not finite?

---

> ### Author Response · Authors · 2023-11-16
>
> We thank the reviewer for their time and constructive feedback on the paper, and are glad to hear they enjoyed reading the paper so much! Please see below for responses on specific queries.
>
> **Formulation of translation-family feature maps.** The translation family of features is a natural choice because it is simple to implement and produces good empirical results (as reported in the experimental results in the paper). These functions are similar to radial basis functions used elsewhere; in particular, in prior work on distributed distributional codes (Sahani & Dayan 2003, Vértes and Sahan 2018, 2019). We also emphasize that other families of feature maps can also be considered, and an empirical comparison across a wider variety of feature maps would be an interesting direction for future work.
>
> **The general sketch and first $m$-moments** \
> We are unsure we understand the question fully, so have given a summary of the theory relating to mean embedding sketches and when they can be related to first $m$-moments below; please let us know if we have misunderstood the query or if you have further questions, however.
>
> For a given collection of feature functions $\phi$, and corresponding Bellman coefficients $B_r$, the corresponding computed sketch values do not necessarily correspond to invertible linear combinations of moments. However, the existing theory referenced (Theorem 4.3 of Rowland et al., 2019), guarantees that _if_ the regression error incurred in Equation (5), defining the Bellman coefficients, is 0, then the sketch must correspond to an invertible linear combination of the first $m$-moments, as the reviewer states. We hope this answers the question, but please let us know if not.
>
> **Why can we write $B_r$ as $C_rC^{-1}$?** This follows since Equation (5) defines a least-squares linear regression problem, which $B_r$ solves. The expression $C_r C^{-1}$ is the analogue of the usual $\hat{\beta} = (X^T X)^{-1} X^T Y$ expression for the least-squares coefficients for the linear model expressed in the form $Y = X \beta + \epsilon$. We thank the reviewer for raising this question, and agree that this derivation can be made clearer; we have added a guide to the derivation to Appendix B.6, and in doing so, corrected a typo in Eqn (7).
>
> **Arbitrary reward distributions**\
> Thank you for raising this: The generalization to arbitrary reward distributions is obtained by modifying the  term $\phi(r+\gamma G)$ in the definition of $C_r$ in Eqn (7) to $\mathbb{E}_{R \sim \rho} [\phi(R + \gamma G)]$, where $\rho$ is the distribution of the immediate reward. The intuition behind this is that the regression target is now the _average_ of the sketch values, over the possible values of the immediate reward. To obtain an algorithmic implementation, this is a one-dimensional integral which can be approximated with standard numerical integration techniques, as mentioned in Appendix C.1 (Sketch DP under conditional independence). Experiments on this version of the sketch Bellman operator are shown in the *DC+Gaussian R* columns of Figure 4, and more thorough evaluations are in panels c and d of Figure 7, and panel b of Figure 8.
>
> Additionally, in settings where possible reward values are not available in advance, dealing with unseen reward values $r$ does not require solving the regression problem from scratch. This is because $C^{-1}$ can be saved, and we only need to compute $C_r$ for unseen values of $r$, which is also very fast; we have added a paragraph to Appendix B.6 to describe the details of this approach.

---

### Author Response · Authors · 2023-11-16
**General response**

We thank the reviewers for their useful comments on our paper. Based on the reviewers' queries and suggestions, we now provide several new empirical results, which we believe add further weight to the computational comparisons between the sketch-based algorithms introduced in this paper, and existing distributional algorithms. Responses to additional reviewer queries are included in individual responses below.

Several reviewers expressed interest in seeing quantitative comparisons of efficiency between Sketch-DQN, QR-DQN, and IQN. To this end, we ran an additional experiment in which we report the mean (±s.d.) rate (per second) at which frames are processed during training in our Atari experiments, with each agent running on a single V100 GPU. These frame-processing times reflect the wallclock time associated with all aspects of the DQN training, including network forward passes for action selection, environment simulation, and periodic gradient updates. These statistics are averaged across all games and seeds.

The results are as follows, using either 201 or 401 sketch features/quantiles, or the default number:
* Sketch-DQN: 201 features: 1326±107; 401 features: 1320±110
* QR-DQN: 201 quantiles (default), 1258±107; 401 quantiles: 1256±106
* IQN: 64 quantiles (default), 1120±90; 201 quantiles: 698±41; 401 quantiles: 400±16

The Sketch-DQN method has the highest frame rate, and QR-DQN has a slightly lower average frame rate. IQN with default 64 quantiles has a slightly lower average frame rate still, and is much lower when the number of quantiles is increased to match the number of predictions made by QR-DQN and Sketch-DQN. This is because the IQN architecture requires one forward pass through the MLP component of the network for each predicted quantile level $\tau \in (0, 1)$. By contrast, the Sketch & QR architectures simply modify the original DQN architecture to produce multiple predictions of sketch values/quantiles from the final hidden layer of the network.

Reviewer RJqf also expressed interest in seeing a comparison between Sketch-DP and algorithms based on imputation strategies (statistical functional dynamic programming; SFDP, Rowland et al., 2019; Bellemare et al., 2023). We have run experiments comparing Sketch-DP with the SFDP algorithm based on expectiles, and have included these results in Appendix D.4 of the revised PDF. In short, the results confirm that there are significant benefits in wallclock run-time that come from Sketch-DP, owing to the fact that the DP updates involve only simple linear-algebraic operations, in contrast to the more involved DP update, using imputation strategies, in SFDP.

---

### Comment · Area_Chair_ZQk9 · 2023-11-21
**Engaging in discussion with the authors**

Dear reviewers, we are approaching the end of the discussion period (Nov 22) with the authors , please read the rebuttal and engage with authors to discuss any further questions/ clarifications you may have,


Many thanks

AC

---

### Author Response · Authors · 2023-11-22
**Additional general response**

Following reviewers' comments, we would like to provide an update on the Atari suite results. We ran a version of Sketch-DQN with an increased number of feature functions (401 rather than 201), which attains performance similar to that of IQN. The frame rate stays almost the same, lowering only slightly from 1326 to 1320 frames per second. The results are presented in Figure 5 of the updated PDF. We did not sweep any other hyperparameters, such as learning rate or training schedules.

---

### Meta-Review · Area_Chair_ZQk9 · 2023-12-11

**Metareview:**

The paper proposes to write distributional bellman equation evolution in the sketched space of Mean embeddings of a translation invariant kernel (random fourier features). This enables doing the bellman updates in the embedding space rather than doing it on particle level matching using an MMD distance for example as done in  MMDDRL [1].

The paper was discussed among authors ,reviewers and AC.

Reviewers thought that this is a nice contribution to DRL but two main concerns were raised : 1) on how this method compares theoretically and empirically to the work of MMD-DRL. Adding more discussions  an empirical comparison of the two approaches would resolve this concern.  2) The performance of the method w.r.t to IQN does not lead to significant improvement or can underperform it. Maybe one approach would be to learn the features in a min-max way so that the learned kernel lead to better improvement for the proposed method.

**Justification For Why Not Higher Score:**

Missing emprical and theoretical evaluation of the links between this method and previous work [1].

**Justification For Why Not Lower Score:**

N/!

---

### Decision · Program_Chairs · 2024-01-16

Reject